# Muscle progenitor specification and myogenic differentiation are associated with changes in chromatin topology

Nan Zhang[1], Julen Mendieta-Esteban [2], Alessandro Magli [3,4], Karin C. Lilja[1], Rita C. R. Perlingeiro [3,4], Marc A. Marti-Renom [2,5,6,7], Aristotelis Tsirigos [1] & Brian David Dynlacht[1✉]

Using Hi-C, promoter-capture Hi-C (pCHi-C), and other genome-wide approaches in skeletal muscle progenitors that inducibly express a master transcription factor, Pax7, we systematically characterize at high-resolution the spatio-temporal re-organization of compartments and promoter-anchored interactions as a consequence of myogenic commitment and differentiation. We identify key promoter-enhancer interaction motifs, namely, cliques and networks, and interactions that are dependent on Pax7 binding. Remarkably, Pax7 binds to a majority of super-enhancers, and together with a cadre of interacting transcription factors, assembles feed-forward regulatory loops. During differentiation, epigenetic memory and persistent looping are maintained at a subset of Pax7 enhancers in the absence of Pax7. We also identify and functionally validate a previously uncharacterized Pax7-bound enhancer hub that regulates the essential myosin heavy chain cluster during skeletal muscle cell differentiation. Our studies lay the groundwork for understanding the role of Pax7 in orchestrating changes in the three-dimensional chromatin conformation in muscle progenitors.

[1] Department of Pathology and Perlmutter Cancer Institute, New York University School of Medicine, New York, NY 10016, USA. [2] CNAG-CRG, Centre for Genomic Regulation (CRG), Barcelona Institute of Science and Technology (BIST), Barcelona, Spain. [3] Department of Medicine, Lillehei Heart Institute, University of Minnesota, Minneapolis, MN 55455, USA. [4] Stem Cell Institute, University of Minnesota, Minneapolis, MN 55455, USA. [5] Centre for Genomic Regulation (CRG), Barcelona Institute of Science and Technology (BIST), Barcelona, Spain. [6] Universitat Pompeu Fabra (UPF), Barcelona, Spain. [7] ICREA, Barcelona, Spain. ✉email: Brian.Dynlacht@nyulangone.org

Chromatin structure and topology play important roles in the regulation of gene expression. Although it is well established that enhancers can activate or repress gene expression through looping to promoters[1–5], the extent to which this phenomenon directs specification of myogenic precursors and differentiation of skeletal muscle has not been extensively investigated. Mammalian adult skeletal muscle has a robust ability to regenerate, a process that depends on muscle stem cells, termed satellite cells. Satellite cells are characterized by expression of the transcription factor (TF) paired-box 7 (Pax7), which plays a central role in satellite cell specification, maintenance, and function[6–8]. Previous studies have shown that Pax7 can act as a pioneer factor to open local chromatin regions and preferentially bind to enhancers in skeletal muscle progenitors and the pituitary[9–11]. However, it remains unclear whether and how Pax7 can control gene expression through local chromatin remodeling, enhancer activation, and long-range interactions.

Whereas topological changes in chromatin have been extensively investigated in diverse tissue and cell types[12–18], technical obstacles have hindered similar discoveries in skeletal muscle progenitors. For example, satellite cells represent a low-abundance population in muscle tissue, and these cells spontaneously differentiate once isolated. We sought to overcome this obstacle through the induced expression of Pax7 in mouse embryonic stem cells (ESCs), from which skeletal muscle progenitors, termed iPax7 cells, were derived. Upon transplantation into dystrophic mice, iPax7 muscle progenitors functionally mimic satellite cells and are able to seed the stem cell niche and ameliorate muscle wasting[19,20]. Importantly, iPax7 progenitors also recapitulate the transcriptomic and epigenetic features of satellite cells[11]. Thus, iPax7 myogenic cells represent a good model for investigating how this TF regulates genomic architecture in muscle progenitor cells.

To assess genome-wide changes during progenitor specification and differentiation in iPax7 cells, we performed Hi-C and promoter capture Hi-C (pCHi-C), which revealed an extensive three-dimensional (3D) reorganization of chromatin. By comparing our results with data from mouse ESCs[21], and aided by 3D modeling of pCHi-C interactions and proteome-wide capture of Pax7-interacting proteins, we identified enhancer hubs (EnHs) and elucidated key promoter–enhancer (P–En) interaction motifs, spatio-temporal rewiring of P–En interactions, and the corresponding impact on gene expression in each cell population during myogenesis. We identified two classes of Pax7-associated P–En contacts, the maintenance of which was either Pax7-dependent or independent. Enhancers from the latter group were associated with recruitment of additional myogenic TFs and epigenetic memory during differentiation, and they retained their long-range interactions and an open and active state upon the loss of Pax7. Furthermore, we show that Pax7 binds most super-enhancers (SEs) that interact with target promoters in iPax7 muscle progenitors and that it establishes feed-forward loops with a cohort of TFs with which it physically interacts, providing an explanation for the pivotal role of this factor in skeletal muscle stem cell specification and maintenance. Lastly, using epigenome editing, we show that a previously uncharacterized Pax7-bound EnH within the myosin heavy chain (Myh) cluster can activate three Myh genes needed to build muscle.

## Results

### Global chromatin conformational changes during muscle cell progenitor specification and differentiation. 
We used mouse iPax7 myogenic progenitors to explore the 3D organization of the genome during skeletal muscle differentiation. In the presence of doxycycline (Dox), iPax7 cells proliferate as muscle precursors, and after its withdrawal, cells differentiate and display the morphology and gene expression profiles of myocytes (Fig. 1a)[11,19]. To characterize how chromatin structure is globally re-organized during skeletal muscle progenitor specification and differentiation, we generated chromosomal conformation maps using in situ Hi-C and pCHi-C and performed chromatin immunoprecipitation (ChIP)-seq to detect CTCF and cohesin (Smc3) recruitment in both progenitor (+Dox) and differentiated (−Dox) iPax7 cells (Fig. 1b) and compared our results to published data from mouse ESCs[22] (Supplementary Data 1). We identified between 2755 and 2841 topologically associated domains (TADs) in undifferentiated ESCs, progenitor, and differentiated iPax7 cells (Supplementary Fig. 1A, B), and >85% of TAD boundaries were bound by CTCF and cohesin (Smc3). As previously reported[22], TAD boundaries remained largely stable across different cell populations (Fig. 1c). However, we found that promoter-anchored inter-TAD interactions (detected by pCHi-C) occurred more frequently in ESCs than in iPax7 skeletal muscle progenitors, and the number of interactions and average interaction distance did not further decrease during iPax7 cell differentiation (Fig. 1d, e and Supplementary Fig. 1C–E). These restrictions on inter-TAD interactions could result from changes in chromatin condensation during the process of lineage specification and cell differentiation[23].

We also used our Hi-C data to explore genome compartmentalization, wherein active (A) and repressed (B) chromatin can be segregated into two compartments[24]. Notably, we observed striking compartmental switching from ESCs to iPax7 progenitors, altering ~20% of the genome. In contrast, during iPax7 cell differentiation, the scale of compartment reorganization was considerably dampened and comprised only ~6.5% of genomic regions (Fig. 1f, g). Overall, changes in the enrichment of active histone modifications and accessible chromatin were consistent with the activity of switched compartments (Supplementary Fig. 1F, G). Notably, genes within compartments undergoing switching were enriched for skeletal muscle development- and cell identity-related annotations (Fig. 1h). Taken together, our results suggest that chromatin conformational changes primarily occur at an early stage during skeletal muscle lineage specification, with additional conformational changes occurring less frequently during myogenic differentiation.

### Demarcating stable and transient P–En interactions during skeletal muscle specification and differentiation. 
To understand how cis-regulatory elements regulate myogenic differentiation and gene expression in adult skeletal muscle cells, we generated high-resolution pCHi-C maps capturing genome-wide, long-range interactions with a curated set of 25,747 Ensembl annotated promoters in iPax7 mouse skeletal muscle progenitors (+Dox) and differentiating myocytes after Dox withdrawal (−Dox) (Fig. 1b). We also compared our results with published data from mouse ESCs[21], from which iPax7 cells originate. After merging data from replicates, we used CHiCAGO[25] to identify 107,330 to 121,802 high-confidence interactions between annotated promoters and distal promoter-interacting DNA fragments in ESCs, Dox-treated, and differentiated conditions (Supplementary Fig. 2A and Supplementary Data 1). The majority (>99%) of captured interactions were found in cis (Supplementary Data 1), and >65% of them linked promoter to non-promoter regions in all three cell populations, with median distances between 131 and 155 kb (Supplementary Fig. 2A–C). In addition, we examined genome-wide RNA-seq data and found that high-confidence pCHi-C interactions were detected more often at promoters of expressed genes (Supplementary Fig. 2D). As expected, promoters and promoter-interacting regions for expressed genes generally

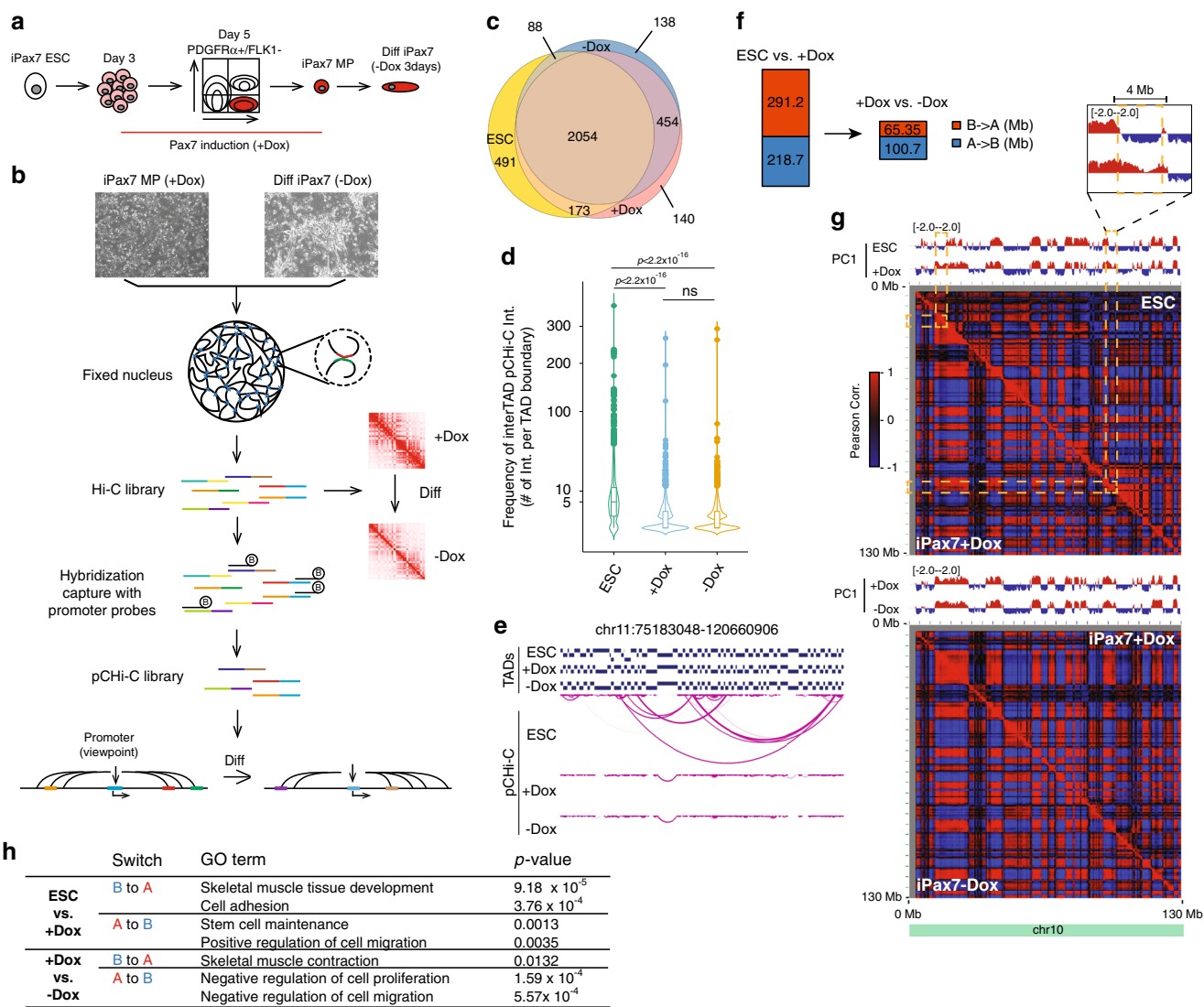

**Fig. 1 Global chromatin conformational changes during muscle cell progenitor specification and differentiation. a** Scheme for the generation of iPax7 skeletal muscle progenitors (MP) and differentiated derivatives (Diff). **b** Scheme showing experimental design to generate genome-wide Hi-C and pCHi-C maps in undifferentiated (+Dox) and differentiated (−Dox) iPax7 cells. **c** Venn diagram showing overlap of TADs identified in ESC, iPax7+Dox, and iPax7−Dox populations. **d** Violin plot showing the frequency of significant inter-TAD interactions identified by pCHi-C at each TAD boundary in ESCs ($n = 2816$) and iPax7 (+Dox ($n = 2821$) and –Dox ($n = 2734$)) cells. The boxplot within each violin plot shows the 25th and 75th percentile (bottom and top of box), and median value (horizontal band inside box). The whiskers indicate the values observed within up to 1.5 times the interquartile range above and below the box. Statistical significance tested with two-tailed Student's $t$-test. **e** Representative example of a genomic region with high-confidence inter-TAD pCHi-C interactions primarily observed in ESCs but not iPax7 cells. Each magenta arc depicts a high-confidence pCHi-C interaction. **f** Quantification of genome-wide compartment switches from B to A (red) and A to B (blue) for ESCs vs. +Dox iPax7 and +Dox vs. −Dox iPax7 populations. Switches are quantified in Mb. **g** Compartment changes at chr10 for ESCs vs. iPax7+Dox (top) and iPax7+Dox vs. iPax7−Dox (bottom). PC1 value tracks (red: positive values; blue: negative values) and Pearson correlation matrices for the intra-chromosomal interaction profiles were generated at 500 kb resolution for comparison. Selected regions showing compartmental switching are highlighted with yellow dashed rectangles. **h** GO analysis of select groups of genes located in compartments altered when comparing ESCs vs. +Dox iPax7 cells and +Dox vs. −Dox iPax7 populations.

showed greater enrichment for active chromatin features than non-expressed genes in ChIP-seq experiments (Supplementary Fig. 2E, F).

To further characterize epigenetic features of promoter-interacting regions, we annotated genome-wide open chromatin regions in ESCs and undifferentiated and differentiated iPax7 muscle progenitors using ATAC-seq and ChIP-seq datasets generated by our lab and others (Supplementary Table 1), resulting in three distinct clusters in iPax7 cells and four clusters in ESCs (Fig. 2a and Supplementary Fig. 3A, B)[11,26]. Based on

previous knowledge[27], we defined active enhancers as accessible, non-promoter regions marked with both H3K4me1 and H3K27ac modifications. We also used our H3K27ac data to define SEs[28]. This compendium of enhancers was enriched with different patterns of histone modifications and CTCF and Smc3 recruitment in ESCs and iPax7 cells (Fig. 2b and Supplementary Fig. 3C, D). In iPax7 muscle progenitors, active enhancers were primarily observed in one cluster (group III) that predominantly contained non-promoter regions and that overlapped extensively with Pax7 occupancy by ChIP-seq. Indeed, these enhancers were highly

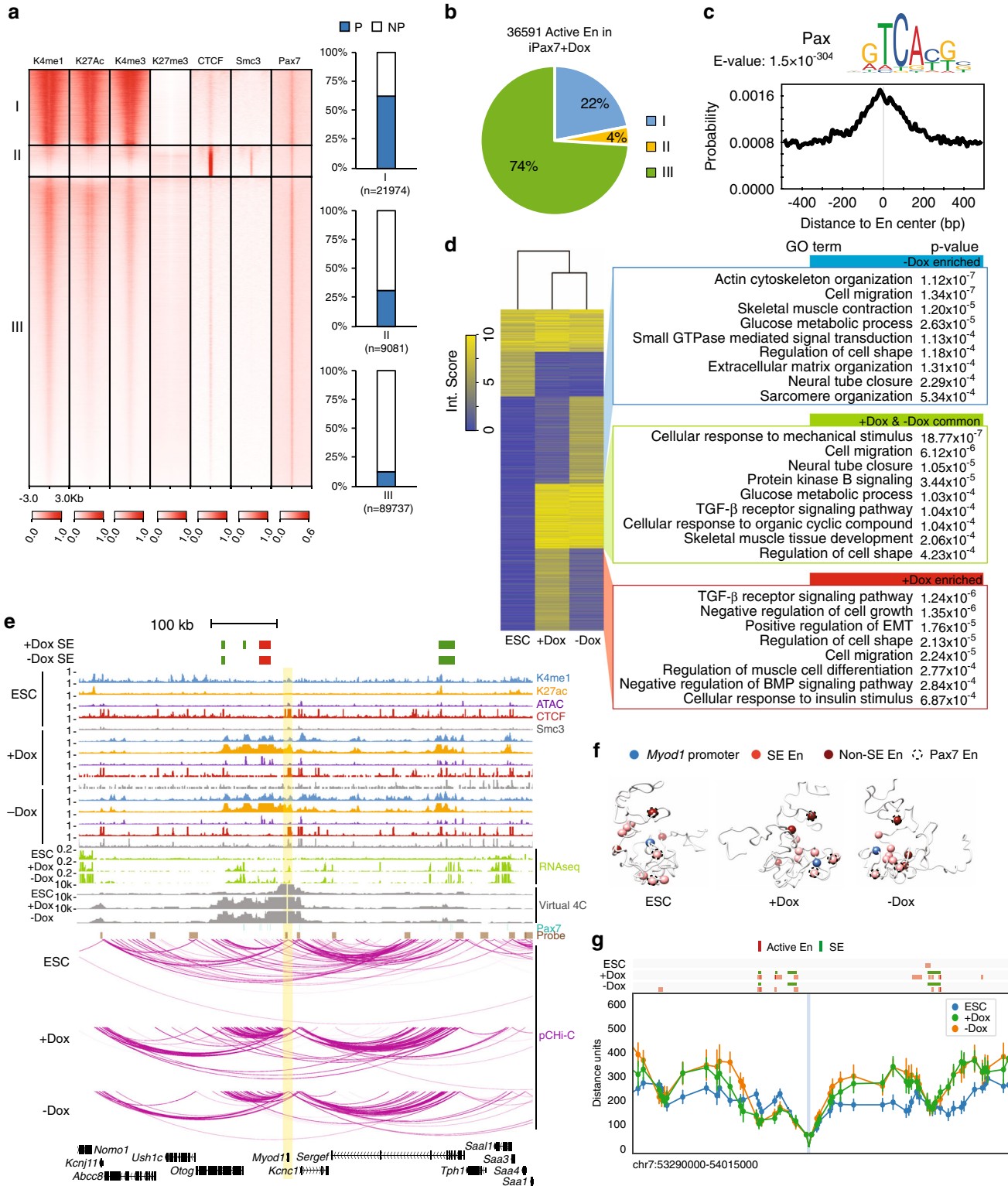

enriched for the paired-box domain binding motif, a well-known Pax7 recognition sequence (Fig. 2c).

We visualized global changes in P–En interactions by plotting heat maps depicting high-confidence interactions in all three populations (Fig. 2d). Most strikingly, interactions between promoters and active enhancers were rewired in a lineage- and differentiation-dependent manner, and iPax7-specific P–En interactions were observed at genes with functional relevance

for skeletal muscle biology (Fig. 2d, Supplementary Fig. 3E, F, and Supplementary Data 2). Interestingly, we observed a cluster of genes ($n = 3019$) that transiently flipped their connections with enhancers in iPax7 cells: although distal enhancers interacted with these target genes in Dox-treated progenitors, these connections were reversed upon Dox removal (Fig. 2d). Many of these genes are involved in TGF-β, insulin, and BMP signaling pathways, as well as epithelial–mesenchymal transition (EMT). In

**Fig. 2 Genome-wide rewiring of P–En interactions during myogenic specification and differentiation. a** Features of open chromatin were demarcated in iPax7 muscle progenitors. Open chromatin regions in iPax7 progenitors (+Dox) were detected by ATAC-seq and then classified with *k*-means clustering on the basis of epigenomic features and co-localization with Pax7, CTCF, and Smc3. Each group of open chromatin regions was further annotated with genic features based on the location of peaks within the group, and the result was presented as a bar plot (right). P, promoter; NP, non-promoter region. **b** Quantification of active enhancers detected in iPax7 progenitors based on grouping in **a**. **c** Binding site motif enrichment of paired-box transcription factor at active enhancers from group III in undifferentiated iPax7 (Dox-treated) cells. **d** Classification of high-confidence P–En interactions in ESCs and iPax7 cells captured by pCHi-C. (Left) High-confidence pCHi-C interactions called by CHiCAGO (interaction score ≥5) from either one of the three cell populations (ESCs, +Dox, and −Dox iPax7) were classified using *k*-means clustering. (Right) GO analysis for genes with interactions enriched in −Dox or +Dox populations and interactions common to both populations. **e** Genome browser tracks encompassing the *Myod1* locus (highlighted in yellow), showing ChIP-seq, ATAC-seq, RNA-seq, Pax7 binding sites, pCHi-C probes, and high-confidence pCHi-C interactions (represented with magenta arcs) and virtual 4C tracks in ESCs and iPax7 cells before (+Dox) and after (−Dox) myogenic differentiation. SEs were only detected in +Dox and −Dox iPax7 cells. SEs containing the "core" *Myod1* enhancer are marked in red. **f** 3D chromatin conformation models of the *Myod1* locus generated from pCHi-C data from ESCs, +Dox, and −Dox iPax7 cells. The top-scoring 3D models for *Myod1* in each population are shown. **g** Distance distribution between *Myod1* promoter (highlighted in blue) and various features (active enhancer and SE as shown on top of the line plot) for each cell population. Line plot at 5 kb resolution displays the median distance distribution between the *Myod1* promoter and all the particles containing a CTCF peak or an active enhancer in the ensemble of models of ESC (blue, *n* = 948), +Dox (green, *n* = 489), and −Dox (orange, *n* = 476). The bar displayed for most of the dots indicates one standard deviation from the distribution median.

contrast, other interactions that were observed in Dox-treated cells persisted after Pax7 expression ceased. These findings suggest that muscle progenitors undergo transient, as well as sustained, changes in chromatin topology to modulate expression of myogenic genes during progenitor specification. Our findings demonstrate the robustness and feasibility of mapping transient and sustained P–En interactions with functional relevance using pCHi-C and provide a resource to describe changes in chromatin topology during myogenic progenitor specification.

**Re-organization of the *Myod1* locus in progenitors**. We sought to reveal key topological changes associated with myogenic differentiation and focused initially on master regulatory proteins associated with this process. MyoD1 is a myogenic regulatory factor (MRF) specifically expressed in skeletal muscle cells during myogenic differentiation[29,30]. Consistently, *Myod1* promoter-anchored interactions were observed primarily at regions that become active enhancers and SEs in iPax7 cells (Fig. 2e). Notably, a "core" enhancer, located ~25 kb upstream of *Myod1* and previously demonstrated to control *Myod1* expression in vivo during myoblast commitment in embryos[31–33], was also found to interact with the *Myod1* promoter in our study. Remarkably, we found that additional enhancers, beyond the "core", comprised SEs that coordinately interacted specifically in muscle cells. In addition to detecting high confidence interactions with CHi-CAGO, we visualized normalized reads from pCHi-C experiments in virtual 4C plots, and the results remain consistent. We also employed our pCHi-C data to develop 3D models of the *Myod1* locus. Accordingly, top-scoring models for this locus suggested that regions that become active enhancers and SEs, including the "core" enhancer, cluster substantially closer to the *Myod1* promoter, as compared to nearby regions (Fig. 2f, g). These results implicate multiple, previously unidentified regulatory inputs that govern muscle-specific expression of *Myod1* through long-range interactions.

Our findings suggested that global promoter interactomes were altered in a myogenic lineage- and differentiation-dependent manner. As a further test, we also compared our pCHi-C results to a second mesodermal differentiation system, namely, adipogenesis. 3T3-L1 pre-adipocytes have been examined by pCHi-C before and after adipogenic differentiation[34]. High-confidence interactions around *Myod1* and a second MRF (*Myog*) as well as adipogenic (e.g., *Pparg*) gene promoters were observed extensively, but also selectively, in their respective cell types (Fig. 2e and Supplementary Figs. 4A, B and 5A). We also found that

tightly interacting Polycomb-associated domains encompassing the HOX clusters in ESCs[35] were significantly altered in both iPax7 and 3T3-L1 cells to enable new interactions within and beyond this cluster (Supplementary Fig. 5B). Our findings thus reveal lineage-specific long-range interactions linked to expression of key muscle regulatory factors in progenitors and their differentiated derivatives (also see below).

**Spatio-temporal alterations in topology and gene expression through assembly of P–En cliques**. During development, enhancers play a critical role in regulating cell type-specific gene expression, which often involves long-range chromosomal interactions with their target genes[1,3–5,36]. Therefore, we sought to further elucidate how P–En interactions were globally rewired as a function of myogenic specification and differentiation. Strikingly, we found that most active enhancers in iPax7 cells (~90%) tended to form an interaction motif that we term a P–En clique, wherein a single promoter was connected to ≥2 active enhancers, and this type of interaction involved ~60% of pCHi-C captured promoters in iPax7 cells (Fig. 3a and Supplementary Fig. 6A).

Importantly, we found that the majority (>96%) of P–En interactions detected in iPax7 cells were not observed in ESCs (Supplementary Fig. 6B). Therefore, to better understand the relationship between rewiring of P–En interactions and muscle cell differentiation, we grouped these high-confidence interactions in iPax7 cells into those that were unique to a given condition ("+Dox unique" and "−Dox unique") or commonly observed in both conditions (Fig. 3b). Consistent with its function and expression profile, Pax7 binding was more highly enriched at interacting enhancers unique to Dox-treated cells, while in striking contrast, TFs that are continuously expressed throughout adult myogenesis or during later stages of muscle progenitor differentiation, such as Runx1, c-Jun, Six4, Foxk1, Tead1, Tead4, and Myod1, were more highly enriched within enhancers that interacted in both progenitors and differentiating cells (Supplementary Fig. 6C).

We segregated all enhancer target genes detected in iPax7 cells into seven groups (I–VII) based on their distinct patterns of condition-specific rewiring of P–En interactions found in these cells during differentiation (Fig. 3c, d and Supplementary Fig. 6D–G). Notably, we observed many instances in which P–En contacts were either completely erased (group I) or established de novo (group III) after Dox removal (Fig. 3c, d). The majority (59%) of promoters targeted by Pax7-bound enhancers belonged to groups I and II, which exhibited either

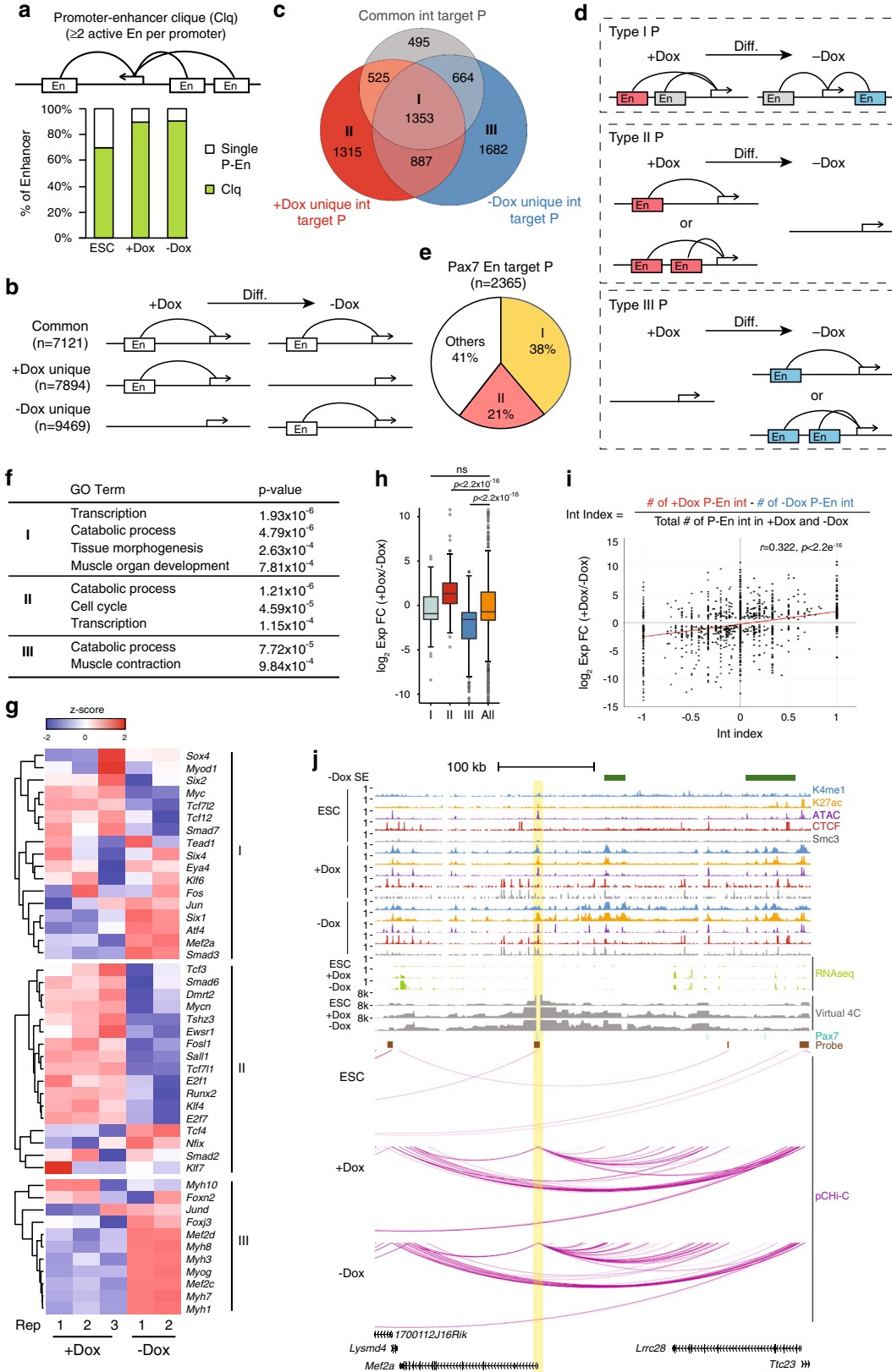

partial or complete loss of specific P–En interactions after Dox removal (Fig. 3e). Genes from these three groups were enriched in muscle tissue development and functionally related terms, and their transcription levels were coordinately regulated (Fig. 3f–h).

Of note, we observed an overall modest, yet positive correlation between gene expression and the number of active enhancers that interact with a given promoter at each specific stage of muscle differentiation (Fig. 3i). For example, consistent with its higher expression level in differentiated iPax7 cells, the promoter of *Mef2a*, a key muscle regulatory gene[37,38], interacted with substantially more enhancers after Dox removal than in Dox-induced cells (Fig. 3j). 3D-modeling of the *Mef2a* locus also

**Fig. 3 Spatio-temporal remodeling of P–En interactions controlling transcription during myogenic differentiation. a** Schematic representation of a P–En clique (Clq, top) and quantification of active enhancers involved in P–En cliques (bottom); $n = 2325$ for ESCs, $n = 9380$ for iPax7 progenitors (+Dox), and $n = 10,452$ for differentiating iPax7 cells (−Dox). In each panel, P and En indicate promoter and active enhancer, respectively. **b** Schematic representation of three types of rewiring of P–En interactions during iPax7 myogenic progenitor differentiation. **c, d** Classification of enhancer-interacting promoters (P) according to the three modes of rewiring in **b** during iPax7 cell differentiation. Promoters were grouped into seven types (labeled I–VII), and the number of promoters found in each type is denoted in the Venn diagram (**c**). For simplicity, schematics of rewiring of P–En interactions during differentiation for type I–III promoters are shown in **d**, with common enhancers in gray, +Dox specific enhancers in red, and −Dox specific enhancers in blue. Type IV–VII promoters are shown in Supplementary Fig. 6 (D, E). **e** Quantification of Pax7 enhancer target promoters showing that most belong to type I and II promoters in **c**. **f** GO analysis of genes with type I–III pCHi-C promoters in **c**. **g** Expression of selected genes with type I–III pCHi-C promoters (from **c**) in undifferentiated (+Dox) and differentiating (−Dox) iPax7 cells. Expression levels for each gene from each replicate are represented by a z-score for normalized RNA-seq data. **h** Transcriptional changes for genes with type I–III pCHi-C promoters grouped in **c** during iPax7 progenitor differentiation. Only differentially expressed genes ($p_{adj} < 0.05$ from DESeq2) are included ($n = 197, 304$, and 388 for types I, II, and III, respectively). The complete set of differentially expressed genes with enhancer interactions during iPax7 cell differentiation was used as a control (All, $n = 2831$). The boxes denote the 25th and 75th percentile (bottom and top of box), and median value (horizontal band inside box). The whiskers indicate the values observed within up to 1.5 times the interquartile range above and below the box; p values result from two-tailed Student's t-test. **i** Positive correlation between enhancer interaction index and gene expression levels of its targets. (Top) Equation to calculate the interaction (Int) index for each enhancer target promoter. (Bottom) Scatter plot showing the distribution of $\log_2$ expression fold-change (FC) during iPax7 cell differentiation as a function of interaction index for genes with P–En interactions captured by pCHi-C in iPax7 cells. A linear regression line ($y \sim x$) is plotted in red; r, Pearson correlation coefficient. Statistical significance given by two-tailed Student's t-test. **j** Genome browser tracks encompassing *Mef2a* locus (promoter is highlighted in yellow), showing ChIP-seq, ATAC-seq, RNA-seq, Pax7 binding sites, pCHi-C probes, and high-confidence pCHi-C interactions (magenta arcs, with corresponding virtual 4C plots) in ESCs and iPax7 cells before (+Dox) and after (−Dox) differentiation. SE is only detected in iPax7 cells without Dox (−Dox).

suggested that distances between most enhancers and the *Mef2a* promoter were significantly reduced in iPax7 myogenic progenitors, as compared to ESCs, and they were further reduced after Dox removal in differentiated myocytes (Supplementary Fig. 7A, B). *Myf5*, another MRF and Pax7 target[39], is likewise integrated into a P–En clique. Although the *Myf5* promoter also contacted an enhancer after Dox removal, iPax7 progenitors exhibited considerably more interactions. Moreover, interactions with the −111 kb enhancer—which is known to be active in embryos[40]— were uniquely detected in Dox-treated cells (Supplementary Fig. 7C). Furthermore, and consistent with this observation, we showed that this enhancer is part of an SE de-commissioned after Dox withdrawal and that *Myf5* is more highly expressed in Dox-treated iPax7 cells.

Taken together, our observations at multiple loci encoding MRFs (*Myod1*, *Myog*, and *Myf5*) and *Mef2a* suggest a role for spatial-temporal rearrangement of individual P–En interactions and P–En cliques in regulating gene expression during muscle progenitor specification and differentiation.

**Epigenetic memory at Pax7 enhancers is established in muscle progenitors.** Given that the myogenic progenitor state is maintained by expression of Pax7, we focused our attention more specifically on Pax7-bound enhancers. Previously, we showed that Pax7 preferentially binds to enhancers and functions locally as a pioneer factor to open and maintain active chromatin in skeletal muscle cells[11]. After improving our ChIP-seq strategy for genome-wide detection of Pax7 recruitment (Methods), we overlaid the resulting 20,579 Pax7 peaks on our pCHi-C data. Unlike previous predictions for enhancer-regulated genes based on nearest TSS, pCHiC results revealed distal targets of Pax7 enhancers that skipped their nearest promoters (Supplementary Fig. 7D). We found that most Pax7-associated enhancer loops were bound by CTCF and cohesin at only one, or neither, of their anchor regions (Supplementary Fig. 7E), suggesting a requirement for additional factors in establishing these long-range loops (see Discussion).

We found that ~90% of Pax7-bound enhancers form P–En cliques, comparable to the percentage observed for all enhancers detected in Dox-treated iPax7 cells (Figs. 3a and 4a). Similar as before, we grouped Pax7-regulated P–En interactions in iPax7

progenitors into two types: those exhibiting (1) interactions that occur in both conditions ("common") versus (2) condition-specific interactions that are detected exclusively in Dox-treated iPax7 myogenic progenitors ("+Dox unique") (Fig. 4b). Consistent with the function of Pax7 as a pioneer factor, we observed a substantial increase in accessibility at these condition-specific Pax7 enhancers in C2C12 myoblasts over-expressing Flag-tagged Pax7 as compared to C2C12 controls. These data indicate that the formation of these enhancers and P–En loops is Pax7-dependent, and they further suggest the possibility that the ability of Pax7 to act as a pioneer factor could coincide with the induction of 3D chromatin interactions.

To investigate the mechanistic basis for the different behavior of enhancers from the two types of Pax7-associated P–En contacts, we explored existing ChIP-seq binding data (Supplementary Table 1) for a group of TFs that are expressed in muscle cells. Remarkably, in contrast with condition-specific enhancers, the enhancers that remained active and maintained interactions after Dox withdrawal showed widespread occupancy by several factors, including MyoD1, Runx1, Six4, Tead1/4, Myog, and c-Jun, in myoblasts (Fig. 4b). These results suggest that a subset of TFs may help maintain enhancer activity and P–En loops during differentiation in the absence of Pax7, subsequent to commitment to the myogenic lineage.

Next, genes that interact with Pax7-bound enhancers were grouped based on their expression patterns during the transition from ESCs to differentiated iPax7 cells (Fig. 4c and Supplementary Data 3). Most of these genes were upregulated in iPax7 progenitors compared to one or both of the other populations. For these active groups ("Active I–III") of genes, we observed enrichment of annotations such as positive regulation of muscle cell differentiation and cell proliferation and skeletal muscle tissue development (Fig. 4d), consistent with previous studies showing that Pax7 ablation leads to progressive loss of satellite cells in newborn mice[6]. We also identified genes (group termed "Primed") that are not highly expressed in iPax7 progenitors but are likely primed by Pax7 enhancers for later activation. In contrast with the active groups, genes from this group were enriched for annotations associated with differentiated muscle, such as negative regulation of cell migration, negative regulation of cell proliferation, and sarcomere organization.

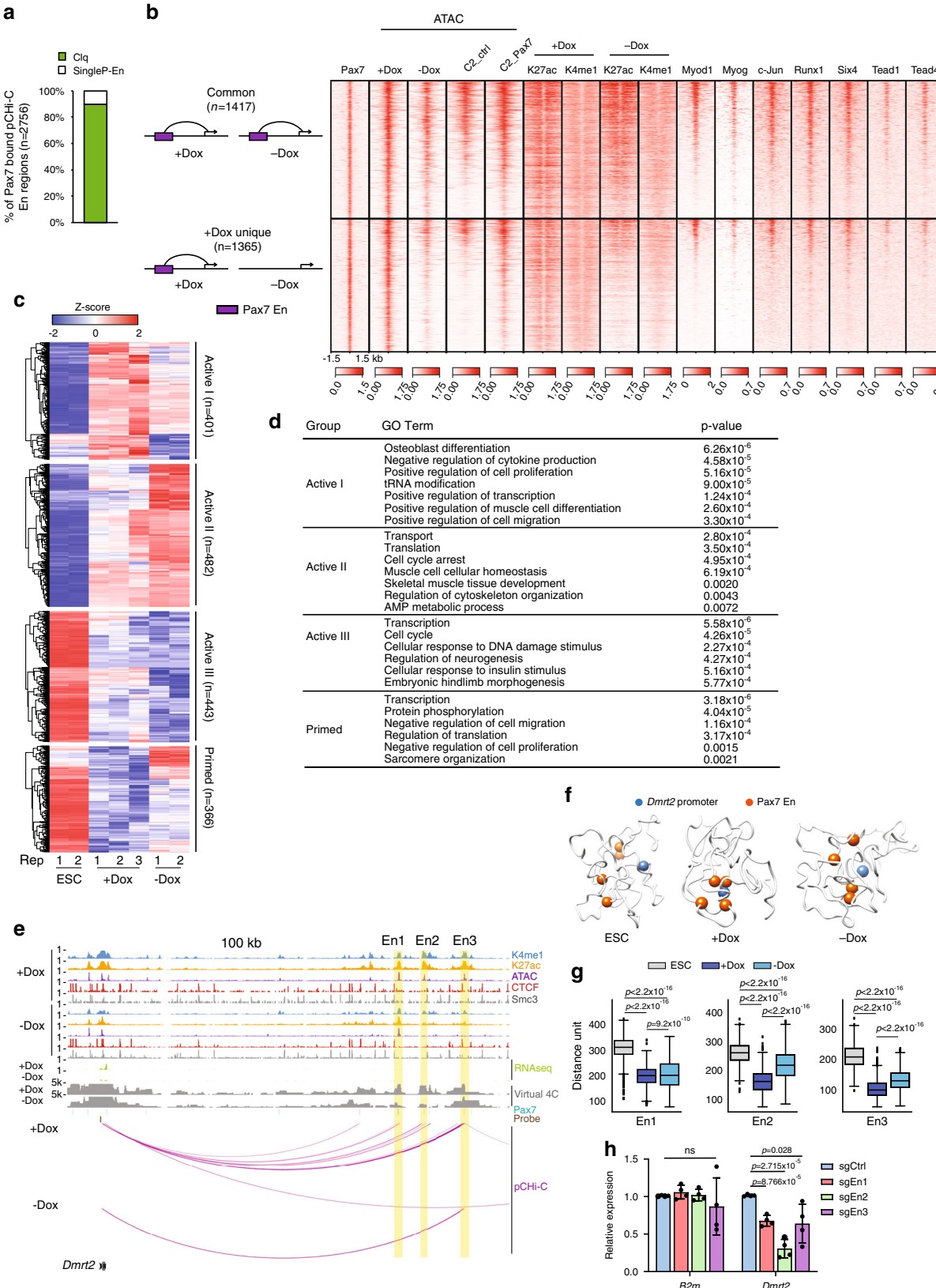

To more closely study the impact of transient interactions with Pax7-bound enhancers, we examined the *Dmrt2* gene, which encodes a TF required for somite maturation and skeletal muscle development that is regulated by the paralog of Pax7, Pax3 (ref. [41]). Here, our results showed that *Dmrt2* was positively regulated through interactions with Pax7-bound enhancers unique to undifferentiated iPax7 cells, wherein it is more highly expressed (Fig. 4e). 3D modeling also indicated aggregation of three Pax7-bound enhancers and the *Dmrt2* promoter in iPax7 progenitors (Fig. 4f, g). To further assess the function of the three Pax7-bound enhancers linked to *Dmrt2*, we used a CRISPR interference (CRISPRi) strategy in which we directed the

**Fig. 4 Two types of Pax7 enhancers collaborate to activate or prime transcription. a** Quantification of Pax7 enhancers involved in P–En cliques (Clq) in +Dox iPax7 cells ($n = 2756$). **b** Promoter-interacting Pax7 enhancers show two distinct patterns of accessibility, histone modifications, and transcription factor binding. (Left) Schematic representation of two types of promoter-Pax7 enhancer (En) interactions. (Right) Heat map of ChIP-seq (TFs and histone modifications) and ATAC-seq signals centered on Pax7 sites (±1.5 kb) suggests preferential co-localization of myogenic transcription factors at Pax7 enhancers involved in common promoter-Pax7 enhancer interactions. C2_ctrl, control C2C12 cells expressing 3xFlag; C2_Pax7, C2C12 cells expressing Pax7_3xFlag. **c** Classification of differentially transcribed genes with Pax7 enhancer interactions detected by pCHi-C, according to their expression levels in ESC and iPax7 populations, as shown. Differentially expressed genes were selected using one-way ANOVA (cut-off $p_{adj}$ of 0.05). Normalized RNA-seq data for each sample were used to calculate $z$-scores. Genes were grouped into four clusters based on the pattern of average $z$-scores across the three cell populations, and each group was further classified with hierarchical clustering. **d** GO analysis for the four groups of genes from **c**. **e** Genome browser tracks around *Dmrt2* locus, showing ChIP-seq, ATAC-seq, RNA-seq, Pax7 binding sites, pCHi-C probes, and magenta arcs representing high-confidence pCHi-C interactions (with corresponding virtual 4C plots) in iPax7 cells before (+Dox) and after (−Dox) differentiation. Yellow, Pax7 enhancer regions (En1–3) that loop to the *Dmrt2* promoter. **f** 3D chromatin conformation models for *Dmrt2* locus based on pCHi-C data. The top-scoring 3D models for *Dmrt2* in ESCs and iPax7 cells are shown. These models show that the three Pax7 enhancers (En1–3) tend to position closer to the *Dmrt2* promoter in Dox-treated iPax7 cells. **g** Distance distributions between the *Dmrt2* promoter and three Pax7-bound enhancers (En1–3), highlighted in **f**, in ESCs and iPax7 cells, obtained from the ensemble of models ($n = 975$) built with the data from each of the cell lines. The boxes denote the 25th and 75th percentile (bottom and top of box), and median value (horizontal band inside box). The whiskers indicate the values observed within up to 1.5 times the interquartile range above and below the box. Statistical significance tested with Kolmogorov–Smirnov test. **h** Relative expression of *Dmrt2* and *B2m* (*β2 microglobulin*) in iPax7 progenitors after CRISPR inhibition with sgRNAs targeting Pax7 sites within *Dmrt2* En1–3; $n = 4$ (biological replicates for each sgRNA). Bars are mean ± s.d., normalized by *Gapdh* and expressed relative to mean levels of the control sgRNA (sgCtrl). Statistical significance tested with two-tailed Student's *t*-test.

repressive dCas9-KRAB-MeCP2 fusion protein to Pax7 sites within each enhancer via specific sgRNAs. Indeed, the expression levels of *Dmrt2*, but not a control gene (*B2m*), decreased after silencing each of these Pax7-bound enhancers (Fig. 4h), suggesting that *Dmrt2* assembles into a functional Pax7-associated P–En clique with contributions from multiple enhancers.

In summary, our studies provide a high-confidence dataset for authentic Pax7 target genes in muscle progenitors. We found two classes of Pax7-associated P–En interactions, in which enhancers are mostly associated with augmentation of gene expression in muscle precursors. Enhancers from one class are decommissioned once Pax7 expression ceases and cells differentiate, as evidenced by their decreased activity and diminished interactions with target promoters. A second group of Pax7-bound enhancers exhibit epigenetic memory associated with retention of enhancer features and persistent looping that are likely maintained by a cohort of TFs expressed in differentiating cells.

**Pax7 assembles extensive feed-forward loops with its target TFs**. To further investigate transcriptional regulatory mechanisms governed by Pax7, we used immuno-affinity purification and mass spectrometric sequencing to identify the compendium of Pax7-associated proteins in muscle progenitors. Although Pax7-associated proteins have been identified in myoblasts[42], it is likely that such an approach would fail to uncover progenitor-specific interactions with this protein. We therefore isolated chromatin from iPax7 cells engineered with a single, inducible copy of the Flag-tagged Pax7 transgene integrated next to the *Hprt* locus (Supplementary Fig. 8A), which is expected to maintain expression levels comparable to those of satellite cells in the presence of Dox. Through purification of Flag-Pax7, we identified a cohort of factors and complexes involved in gene activation or repression and remodeling of chromatin and genome architecture that were substantially enriched compared to the uninduced control (Fig. 5a, Supplementary Fig. 8B, and Supplementary Data 4). We also identified a large cohort of sequence-specific TFs, including several that were shown to play an essential role in muscle stem cells (e.g., Foxk1, Six1, Runx1, Tead1, Nfix)[43–47] (Fig. 5a) and that are recruited to Pax7-bound enhancers in muscle cells (Fig. 4b). Importantly, we also confirmed multiple interactions from our proteomic screen through immunoprecipitation and western blotting (Fig. 5b).

The identification of a cadre of TFs that interacted with Pax7 and that were enriched at Pax7-associated enhancers prompted us to more closely inspect the relationship between these factors by comprehensively merging our proteomic data with the results of ChIP-seq and pCHi-C. Strikingly, we found that many (23 out of 68) of the genes encoding these Pax7-interacting proteins, including *Eya4*, *Jun*, *Dmrt2*, *Myf5*, *Cebpb*, *Runx1/2*, *Six1/2*, *Tcf12*, and *Tead1/4*, were themselves identified as targets of Pax7 enhancers in our pCHi-C experiments. Since several of these factors are able to bind within proximity to Pax7 sites (e.g., Runx1, c-Jun, and Tead1/4 in Fig. 4b), our data suggest that these TFs assemble feed-forward regulatory loops with Pax7 through physical interactions with this protein at enhancers (Fig. 5c). Additionally, a subset of these factors are bound to Pax7 enhancers with persistent chromatin accessibility and looping in differentiating muscle cells after Pax7 expression ceases (Fig. 4b), further supporting our hypothesis that robust recruitment of collaborative, interacting TFs to these enhancers can maintain epigenetic memory.

**Pax7 regulates SE activity in muscle progenitors**. It has been shown that SEs play a role in driving lineage- and development-specific gene expression[48,49]. To systematically investigate the role of SE in regulating myogenic differentiation, we took advantage of H3K27ac ChIP-seq data in ESCs and iPax7 cells to call SEs (Supplementary Fig. 8C and Methods). We found that ~80% of SEs looped to promoters in iPax7 cells (Supplementary Fig. 8D). In contrast with individual active enhancers, SEs showed a positive correlation between the intensity of active chromatin marks and the number of interactions with promoters (Fig. 5d, Supplementary Fig. 8E–G). Further, 61% of promoter–SE interactions detected in Dox-treated iPax7 cells were maintained during differentiation (Supplementary Fig. 8H).

Interestingly, 82% of SEs that interact with target promoters in iPax7 myogenic progenitors contained individual Pax7-bound enhancers (Fig. 5e). It is known that SEs are collectively bound by an array of TFs in different cell types[28,48,50]. In our myogenic progenitors, we detected significant enrichment of motifs and binding sites (by ChIP-seq) for muscle-related TFs, including Jun/Fos, bHLH, Tead, Six, and Runx family members, at individual enhancers within SEs comprised of a Pax7 enhancer (Fig. 5f, g). Notably, factors such as Runx1 and Tcf12 were recruited to nearly all (>90% and ~85%, respectively) of the Pax7-

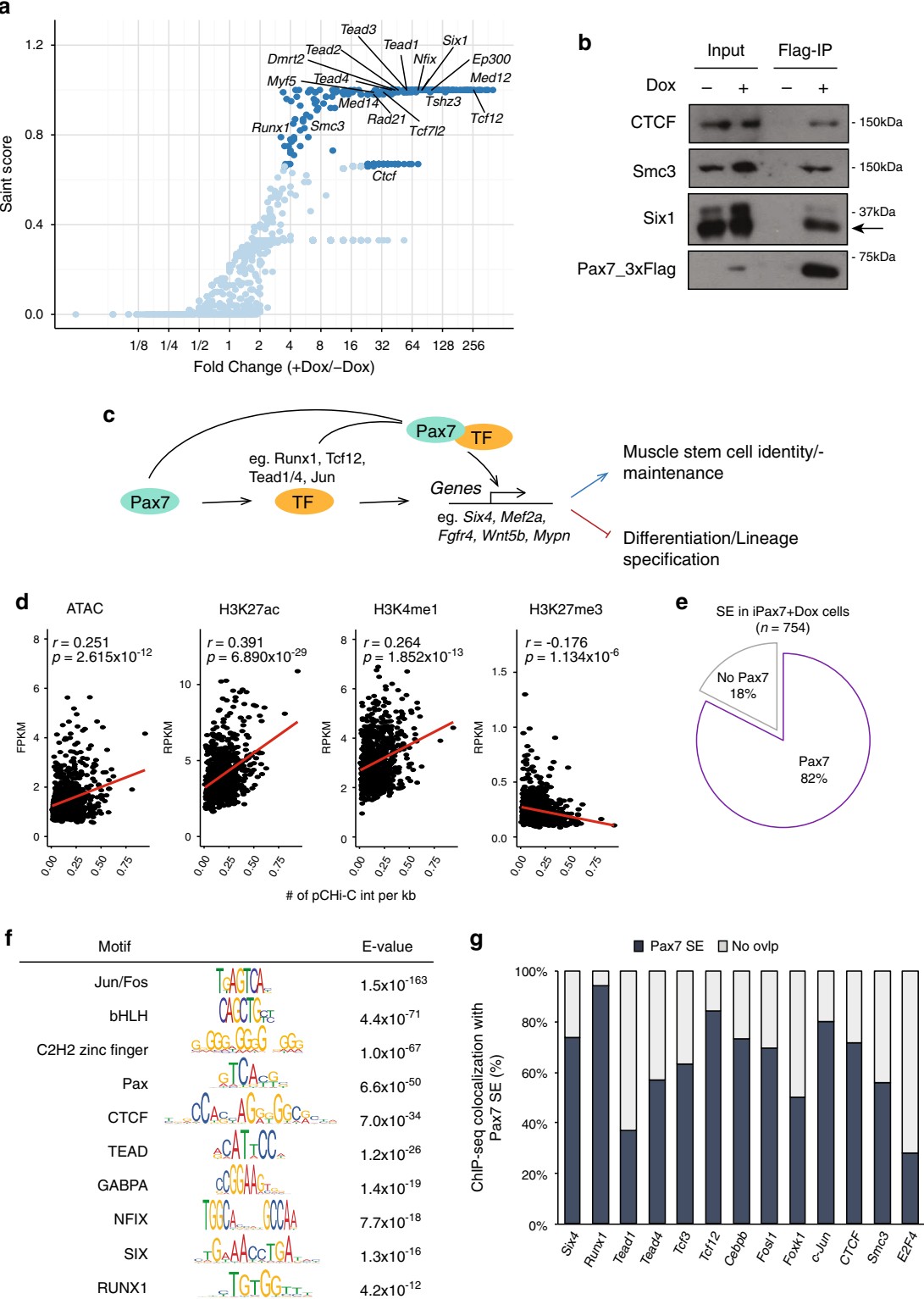

**Fig. 5 Pax7 assembles feed-forward loops with its target TFs. a** SAINT analysis revealing high-scoring Pax7-associated proteins detected by IP-MS. Proteins identified as high-confidence interactors of Flag-tagged Pax7 are marked in dark blue; $n = 3$ (biological replicates). **b** Immuno-blot confirming interactions between indicated proteins and Flag-tagged Pax7 in iPax7 cells. **c** Schematic representation of a feed-forward regulatory loop assembled by Pax7 and its target TFs. **d** Pairwise Pearson correlation between the number of high-confidence pCHi-C interactions per kb and signals for ATAC-seq and ChIP-seq for histone modifications at SEs in iPax7 muscle progenitors. A linear regression line ($y \sim x$) is plotted in red; $r$, Pearson correlation coefficient; $p$ values determined by two-tailed Student's $t$-test. **e** Venn diagram indicates that most SEs identified in +Dox iPax7 cells that display high-confidence pCHi-C interactions contain Pax7-bound active enhancers. **f** Highly enriched TF binding motifs identified in SEs with pCHi-C interactions in +Dox iPax7 cells. The $E$-value is the $p$ value derived from a one-tailed binomial test multiplied by the number of motifs in the input database. **g** Quantification of ChIP-seq peaks detected at Pax7 site-containing SEs. E2F4, control TF not detected as a Pax7 interactor.

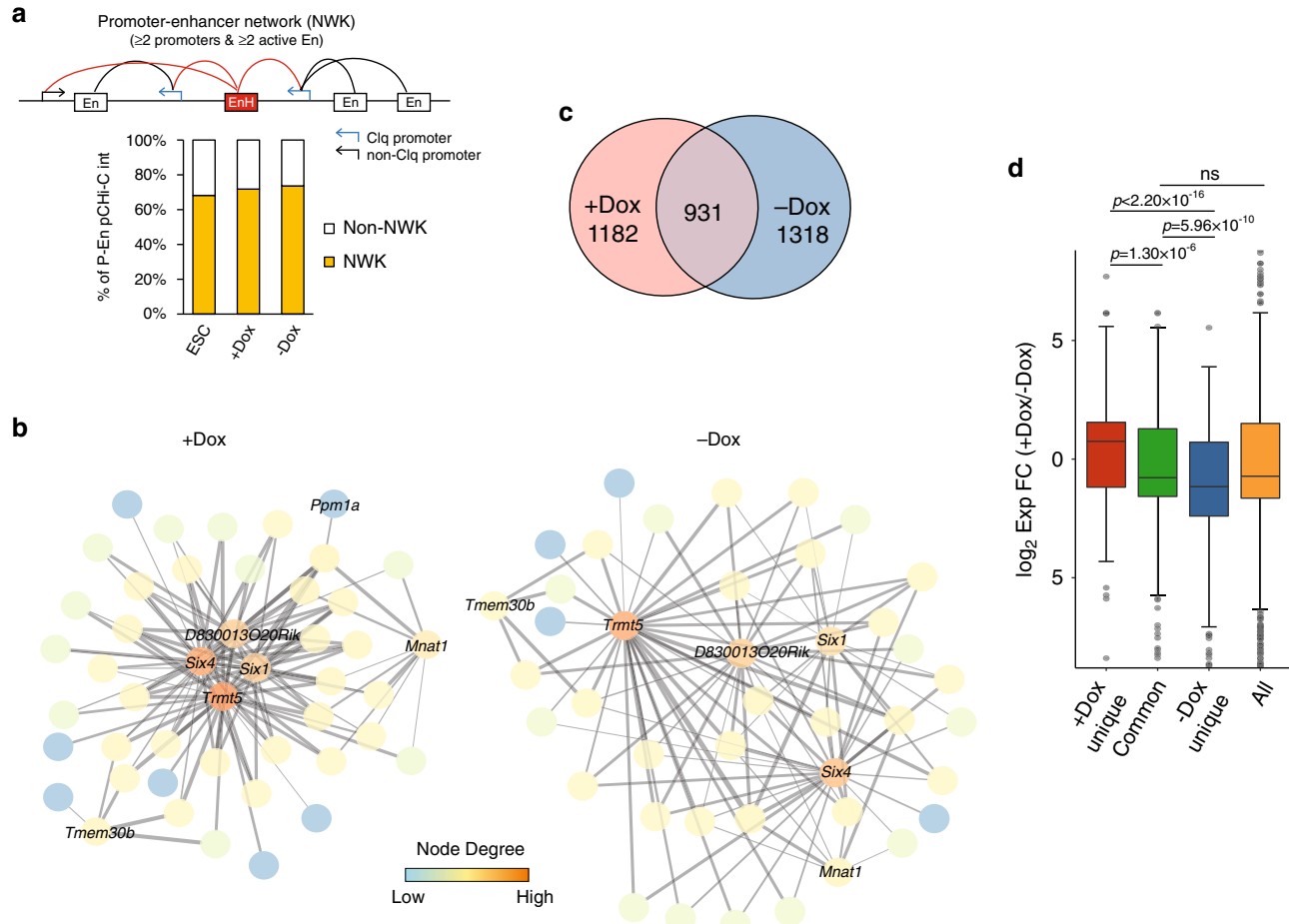

**Fig. 6 A structural and functional role for enhancer hubs (EnHs) in forming P–En networks during myogenic differentiation. a** Identification of P–En networks (NWK) in ESCs ($n = 261$), +Dox ($n = 790$), and −Dox ($n = 880$) iPax7 cells. (Top) Schematic of a P–En network connected through an EnH (in red). (Bottom) Quantification of P–En interactions involved in P–En networks in each population. **b** P–En networks of the *Six1/4* locus in +Dox and −Dox iPax7 cells graphed with Cytoscape. No network was detectable at these loci in ESCs. Each node represents a pCHi-C captured region, and promoters were labeled with their corresponding gene names, while enhancers were left unlabeled. Each gray line between nodes represents a high-confidence pCHi-C interaction, and the thickness of the line indicates the strength of the interaction measured by the CHiCAGO score. Node degree from low to high reflects the relative frequency of connections from one node to other nodes within the motif. **c** Overlap between EnHs in +Dox and −Dox iPax7 cells. **d** Corresponding transcriptional changes during iPax7 cell differentiation for target genes in each EnH group from **c**. Only differentially expressed genes ($p_{adj} < 0.05$ from paired DESeq2) are included ($n = 446, 430,$ and $547$ for group '+Dox unique', 'Common', and '−Dox unique', respectively). All differentially expressed genes in iPax7 cells ($n = 2831$) are used as a control. The boxes denote the 25th and 75th percentile (bottom and top of box), and median value (horizontal band inside box). The whiskers indicate the values observed within up to 1.5 times the interquartile range above and below the box. Statistical significance tested with two-tailed Student's *t*-test.

bound SEs (Fig. 5g). Interestingly, several of these TFs (eg. Runx1, Tcf12, Cebpb, and Tead1/4) physically associate with Pax7 and are also involved in potential feed-forward loops assembled by Pax7 (Fig. 5c). By comparison, factors such as E2F4, which did not physically interact with Pax7, exhibited relatively low enrichment at Pax7-bound SEs (Fig. 5g). Taken together, our data suggest that Pax7 may collaborate with one or more of these TFs to robustly assemble SEs that facilitate and maintain long-distance looping and activation of critical target genes in muscle.

**Enhancer hubs connect P–En cliques to regulate transcription during muscle progenitor specification and differentiation.** Given the complexity of promoter interactomes that we observed during specification and differentiation of iPax7 progenitors (Fig. 3), we further investigated the extent to which these enhancers were integrated into highly connected networks. We found that active enhancer regions tended to

interact with more promoters as compared to regions lacking an enhancer signature (Supplementary Fig. 9A, B). We designated enhancers that interact with ≥2 promoters as EnHs, and found that 33–38% of active enhancers can be classified as such in ESCs and iPax7 cells (Supplementary Fig. 9C). Importantly, in most cases, nearby P–En cliques are interconnected through high-confidence, EnH-mediated interactions. We therefore merged EnH-connected cliques together with EnH target promoters outside cliques, yielding an even more highly connected motif that we term a P–En network (Fig. 6a). In total, we identified 261, 790, and 880 P–En networks in ESCs, +Dox, and −Dox iPax7 cells, respectively. Remarkably, these networks comprise most of the high-confidence P–En interactions in all three cell populations. A network identified specifically in iPax7 cells that involved *Six1* and *Six4*, two TFs with key roles in muscle[51,52], was particularly noteworthy (Fig. 6b). Indeed, both genes assembled into highly connected hubs within each network before and after iPax7 cell differentiation, but not in

ESCs, indicating that the two genes were tightly regulated through enhancer looping in muscle cells.

Interestingly, we found that CTCF binding was significantly enriched at EnHs as compared to enhancers not involved in such hubs (Supplementary Fig. 9D), and this difference was most striking in iPax7 cells undergoing differentiation. These findings suggest a potential role for CTCF at EnHs, although its function at these hubs, if any, remains to be determined. Like other enhancers, EnHs show lineage-specificity, as we detected minimal overlap between hubs in ESCs and either of the two iPax7 populations (Supplementary Fig. 9E). Despite the relatively large percentage of stable EnHs observed during differentiation, more than half of the EnHs were uniquely identified in one of the two iPax7 populations, suggesting that formation of EnHs is also differentiation-dependent (Fig. 6c). As expected, genes controlled by differentiation-dependent EnHs tend to be more differentially regulated compared to genes with promoters connecting to common EnHs (Fig. 6d). Many Pax7 sites were detected at EnHs, with the majority found at those that are maintained during iPax7 cell differentiation (Supplementary Fig. 9F, G), although future studies will be required to assess whether Pax7 functions differently at these EnHs versus other Pax7 enhancers that do not act as regulatory hubs.

Our findings thus reveal considerable genome rewiring during muscle progenitor differentiation, wherein enhancers and genes encoding many key muscle regulators assemble, in a lineage- and differentiation-dependent manner, into highly complex networks that integrate EnHs and cliques.

**Epigenome editing of an EnH that controls expression of multiple *Myh* genes.** Among all EnHs detected in iPax7 cells, we found one of particular interest. The *Myh* cluster encompasses multiple myosin heavy chain (*Myh*) genes, including the developmentally related *Myh3* and *Myh8* genes involved in sarcomere assembly and mutations in which are implicated in muscle diseases[53,54]. This locus is highly conserved between human and mouse genomes with respect to gene order, orientation, and spacing (Fig. 7a). We found that, in differentiating iPax7 cells, this regulatory EnH contacts promoters of the *Myh1*, *Myh3*, and *Myh8* genes that are all upregulated during iPax7 differentiation (Fig. 7a). Further, ChIP-seq results showed that this EnH region contains binding sites for Pax7, Runx1, and Six4, suggesting the possibility that these proteins contribute to the activity of this EnH (Fig. 7b).

We sought to test the functionality of this EnH by interfering with factor binding at this locus through CRISPRi, as before. We showed that each of three sgRNAs that targeted the *Myh* EnH was able to specifically reduce expression from the three *Myh* genes in differentiating iPax7 cells (Fig. 7c). In contrast, expression was not decreased at a control (*Myod1*) locus that was not the target of this EnH. These findings suggest that, indeed, this distal element functions as an important regulatory hub to control expression of requisite structural proteins during muscle differentiation.

**Discussion**
Our work significantly advances the area of epigenetic control of muscle progenitor specification in several ways. First, using Hi-C and pCHi-C data, we found that compartments and inter-TAD interactions were radically altered as pluripotent cells are restricted to myogenic precursors. Additionally, we have mapped genome-wide long-range interactions with promoters in a cellular model recapitulating features of adult muscle stem cells. These datasets represent a unique resource to explore genome-wide interactions between promoters and other *cis*-regulatory elements

and their rewiring during myogenic progenitor specification and muscle differentiation. This approach allowed us to identify complex P–En motifs, including highly connected cliques and networks that are involved in transcriptional control of key players in the specification of progenitors and their differentiation, and provided a high-confidence set of bona fide Pax7 target genes in progenitors. Ultimately, this information may help improve the derivation of muscle stem cells and suggest mechanisms to prevent their disappearance owing to aging and/or wasting.

**A model for Pax7-driven topological rewiring in muscle progenitors.** Interestingly, we found that most SEs contain individual Pax7 enhancers in muscle progenitors, and this finding may in part explain why this factor is essential for specification and maintenance of the satellite cell population in adult skeletal muscle[6,7]. We found that Pax7 binding to a subset of enhancers resulted in sustained interactions, whereas other interactions were readily lost upon iPax7 cell differentiation (Fig. 4b). These observations lead us to propose a model in which the occurrence of such sustained interactions requires recruitment of additional TFs to active enhancers and SEs to preserve epigenetic memory (Fig. 7d). Our conclusion rests on several observations. First, we found that a cohort of sequence-specific TFs were recruited to stable Pax7 enhancers, and such recruitment coincided with the maintenance of looping and an active enhancer signature in differentiating muscle cells (Fig. 4b). Importantly, several of these factors (including MRFs, c-Jun, Six4, and Tead1/4) have been shown to play critical roles in muscle stem cells and myogenic differentiation[51], yet their role in long-range interactions and maintenance of active enhancers has not been documented. Interestingly, the binding of these factors was also highly enriched at SEs that were bound by Pax7 in Dox-treated progenitors (Fig. 5f, g). Second, our proteomic screen detected robust interactions between Pax7 and many of these TFs in iPax7 progenitors. Third, many of the genes encoding these Pax7-interacting TFs were also identified as targets of Pax7 enhancers in our pCHi-C experiments, and both Pax7 and this group of TFs bind enhancers that contact common promoters (Fig. 4b), indicative of Pax7-driven feed-forward regulatory loops (Fig. 5c). Fourth, only a small percentage of Pax7-associated P–En loops were bound by CTCF and cohesin at both loop anchors (Supplementary Fig. 7E), despite the fact that both factors physically interact with Pax7 (Fig. 5a, b). Therefore, it is less likely that these loops were directly mediated through CTCF and/or cohesin. Fifth, another group of Pax7-bound enhancers shows transient looping to promoters in Dox-treated cells—a state reversed after Pax7 removal—and this group of enhancers shows considerably reduced recruitment of TFs such as MRFs, Runx1, Six4, and Tead1/4 (Fig. 4b). Taken together, we hypothesize that more persistent contacts between Pax7 enhancers and promoters may be primarily mediated through these proteins and related Pax7-interacting TFs (Fig. 7d). The recruitment of a cohort of TFs could explain why some enhancers retain epigenetic memory and robust long-range interactions versus those that interact transiently with target promoters, although additional unknown mechanisms could be required to further distinguish these two classes of Pax7 enhancers. Additionally, we note that a cohort of transiently interacting enhancers regulate genes involved in critical signaling cascades during muscle cell differentiation (Fig. 2d), suggesting the need to regulate the topology of P–En interactions within a sharply defined temporal window. In future studies, it will be interesting to determine the phenotypic impact of prolonging or curtailing interactions that are normally transient or persistent, respectively.

 

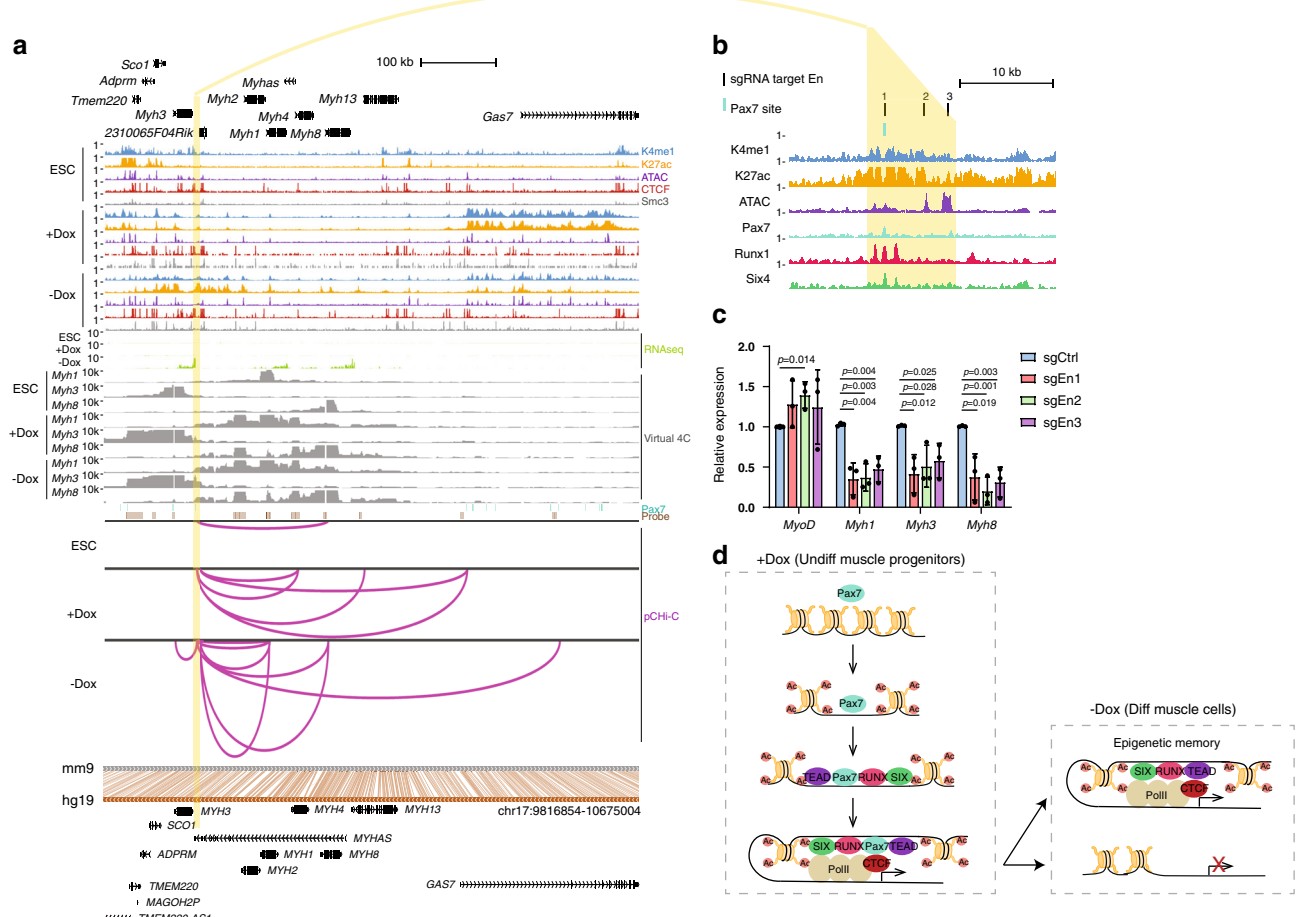

**Fig. 7 An EnH in differentiated iPax7 muscle cells regulates expression of three Myh genes. a** Genome browser tracks (top) around an EnH (highlighted in yellow) identified in iPax7 cells within the *Myh* gene cluster, showing ChIP-seq, ATAC-seq, RNA-seq, Pax7 binding sites, pCHi-C probes, and high-confidence pCHi-C interactions (magenta arcs with associated virtual 4C plots) in ESCs and iPax7 cells before (+Dox) and after (−Dox) differentiation. Bottom, conservation between mouse (mm9) and human (hg19) genomes. Conserved regions are connected with brown lines between aligned genomes. Below hg19 track is the Refseq annotation for human genes at the conserved regions. **b** Transcription factor binding at the *Myh* EnH (yellow). **c** Relative expression of *Myh* EnH target genes in differentiated iPax7 cells after CRISPR inhibition (CRISPRi) with three distinct sgRNAs targeting the EnH; $n = 3$ (biological replicates for each sgRNA). Bars are mean ± s.d., normalized by *Gapdh* and expressed relative to mean levels of the control sgRNA (sgCtrl). Statistical significance tested with two-tailed Student's *t*-test. **d** A model for the re-organization of Pax7-associated P–En interactions during myogenic differentiation. See text for details.

It was shown that Pax3, a Pax7 paralogue, interacts with LIM domain-binding protein 1 (Ldb1) to create long-range loops in cells that represent paraxial mesoderm, yet we did not identify Ldb1 as a Pax7-associated factor in our screen[55]. Interestingly, the majority of Pax7-interacting sequence-specific TFs (60 out of 68) were not identified as Pax3-interacting proteins[55], suggesting that they may be unique interactors for Pax7 in adult progenitors and that Pax7 and Pax3 could employ distinct mechanisms of gene activation, consistent with the ability of Pax7 to specify distinct progenitor populations and to bind to distinct locations by ChIP-seq (Supplementary Fig. 8I). This may also be consistent with the observation that although these Pax proteins bind to overlapping sequences[56] through their paired box and paired-type homeo-domains, they are distinguished by additional, unique domains that enable other co-factors to function at distinct stages of muscle specification.

**Complex, highly connected P–En motifs involved in muscle progenitor specification.** Our pCHi-C experiments revealed layers of complexity involving P–En interactions in muscle precursors. For example, we identified widespread, lineage- and differentiation-dependent formation of P–En cliques and higher order networks connected through EnHs (Figs. 3a and 6a). We showed that such intricate networks are found at the loci of critical myogenic regulators, namely, *Six1* and *Six4* (Fig. 6b). It is likely that the high density of P–En interactions within these networks provides additional regulatory robustness, potentially buttressing against changes in gene expression in the face of the loss of any individual P–En interaction. Of note, we identified a functionally important EnH within the *Myh* cluster (Fig. 7a, b), an evolutionarily conserved cohort of genes essential for muscle function. It will be interesting to further investigate the integrity of networks and the subsequent impact on transcription and skeletal muscle specification and differentiation after disrupting key connecting EnH(s) defined in our study.

## Methods

**Cell culture.** Mouse iPax7 and iPax7_3xFlag skeletal muscle progenitor cells were generated as previously described[19,57,58]. In brief, inducible mouse ESCs were generated by Cre-loxP-mediated recombination of a p2lox-Pax7 or a p2lox-Pax7_3xFlag targeting plasmid into A2lox-cre mouse ESCs[57]. The recombination cassette, located next to the *Hprt* gene, contains the tet-responsive-element (TRE) driving the expression of one single copy of cDNA, thus ensuring quasi-

physiological expression levels. Mouse ESCs were maintained on mitotically impaired mouse embryonic fibroblasts (MEFs) in knockout DMEM (Invitrogen) supplemented with 15% FBS (Embryomax ES-qualified FBS—Millipore), 1% penicillin/streptomycin (Invitrogen), 2 mM Glutamax (Invitrogen), 0.1 mM non-essential amino acids (Invitrogen), 0.1 mM β-mercaptoethanol (Invitrogen), and 1000 U/ml LIF (Millipore). For skeletal myogenic differentiation[58], the ESC/MEF suspension was pre-plated in a gelatin-coated dish for 30 min in order to remove fibroblasts and the resulting supernatant (enriched for mouse ESCs) was then diluted to 40,000 cells/ml in embryoid body (EB) differentiation medium and incubated in an orbital shaker at 80 r.p.m. EB differentiation medium: IMDM (Invitrogen) supplemented with 15% FBS (Embryomax ES-qualified FBS), 1% penicillin/streptomycin (Invitrogen), 2 mM Glutamax (Invitrogen), 50 μg/ml ascorbic acid (Sigma-Aldrich), 4.5 mM monothioglycerol (MP biomedicals). Transgene induction was achieved by adding Dox (Sigma-Aldrich) to day-3 EBs cultures (final concentration 1 μg/ml), and then maintained throughout the differentiation period by replacing the media (including dox) every 2 days. At day 5, EBs were disaggregated and single cells were incubated for 20 min with PDGFRα-PE and FLK1-APC conjugated antibodies (e-Bioscience). PDGFRα+FLK1− cells were sorted using FACSAriaII (BD biosciences) and re-plated on gelatin-coated dishes using EB differentiation media supplemented with 1 μg/ml Dox and 10 ng/ml mouse basic-FGF (Preprotech). Cells were expanded using the same media with Dox for 4 additional days before harvesting them for analysis. iPax7 and iPax7_3xFlag muscle progenitors were grown on 0.1% gelatin-coated culture plates in GlutaMAX supplemented IMDM (Gibco) with 15% stem cell qualified fetal bovine serum (Gemini), 1% penicillin/streptomycin (Corning), 200 μg/ml bovine holo-transferrin (Sigma), 50 μg/ml L-ascorbic acid (Sigma), 4.5 mM 1-thioglycerol (Sigma), and 5 ng/ml recombinant mouse bFGF (R&D systems). Cells were treated with 0.75 μg/ml Dox (Sigma) for the expression of Pax7 or Pax7_3xFlag and were assessed 3 days after removing Dox from the culture media for differentiation.

### Preparation of solubilized chromatin fraction and immune-purification (IP).
Solubilized chromatin fractions from +Dox and −Dox iPax7_3xFlag cells were prepared as described[59]. Briefly, cell pellets were resuspended in buffer A [10 mM Hepes pH 7.9, 10 mM KCl, 1.5 mM MgCl₂, 0.34 M sucrose, 10% glycerol, 1 mM dithiothreitol (DTT), and protease inhibitors (aprotinin, leupeptin, pepstatin A, and phenylmethyl sulfonyl fluoride)], supplemented with Triton X-100 (0.1%), and incubated on ice for 5 min. The nuclear pellet was separated from the cytoplasmic fraction, washed twice with buffer A, and collected by centrifugation. Chromatin-bound proteins were released using buffer A plus 1 mM CaCl₂ and 0.2 U of micrococcal nuclease (Sigma). After incubation at room temperature for 30 min, the nuclease reaction was stopped by the addition of 1 mM EGTA and the nuclear pellet was collected by low-speed centrifugation. Nuclei were then lysed with buffer B [3 mM EDTA, 0.2 mM EGTA, 1 mM DTT, and protease inhibitors (aprotinin, leupeptin, pepstatin A, and phenylmethyl sulfonyl fluoride)]. The chromatin fraction was separated from nucleoplasm by centrifugation at 2250×g for 4 min at 4 °C, washed twice with buffer B, and collected by centrifugation. For immuno-precipitations, the chromatin fraction was resuspended in binding buffer (20 mM Hepes pH 7.9, 100 mM KCl, 0.2 mM EDTA, 20% glycerol, 0.5 mM DTT, and 0.5 mM AEBSF); 1 mg of protein was immunoprecipitated by incubating with mouse anti-FLAG M2 affinity gel (Sigma) at 4 °C for 4 h, and the captured complexes were washed three times with buffer (50 mM Hepes pH 7.9, 250 mM NaCl, 5 mM EDTA, 0.5% NP-40, and 10% glycerol) prior to SDS–PAGE and immuno-blotting. For mass spectrometric sequencing, each IP sample was eluted with 3xFlag peptide and analyzed individually by LC-MS.

Antibodies used for western blotting included: anti-CTCF (Cat. 07-729, Millipore; dilution: 1:2000), anti-Smc3 (Cat. ab9263, abcam; dilution: 1:2000), anti-Flag (Cat. F7425, Sigma; dilution: 1:1000), anti-H3 (Cat. ab1791, abcam; dilution: 1:1000), anti-Pax7 (Developmental Studies Hybridoma Bank (DSHB); dilution: 1:500), and anti-Six1 (Cat. 10709-1-AP, Proteintech; dilution: 1:500).

### Mass spectrometric sequencing
*Sample preparation.* Three biological replicates were prepared for each condition. For each replicate, proteins were reduced with 2.5 μl of 0.2 M dithiothreitol (Sigma) for 1 h at 57 °C at pH 7.5. After samples cooled to room temperature, they were alkylated with 2.5 μl of 0.5 M iodoacetamide (Sigma) for 45 min at room temperature in the dark. NuPAGE LDS Sample buffer (1×) (Invitrogen) was added to the samples then transferred to a NuPAGE 4–12% Bis-Tris Gel 1.0 mm × 10 well (Invitrogen) for SDS PAGE gel electrophoresis. Gel was stained with GelCode Blue Stain Reagent (Thermo Scientific). Sample lanes were excised and destained with 1:1 (v/v) methanol and 100 mM ammonium bicarbonate at 4 °C with agitation. Destained gel pieces were dehydrated in a SpeedVac concentrator. Dried gel pieces were resuspended in 300 μl of 100 mM ammonium bicarbonate with 250 ng Promega trypsin for overnight digestion. A 300 μl solution of 5% formic acid and 0.2% trifluoroacetic acid (TFA) R2 50 μm Poros (Applied Biosystems) beads slurry in water was added to the gel pieces before returning the samples to the shaker for an additional 3 h at 4 °C. Beads were loaded onto equilibrated C18 ziptips (Millipore), with 0.1% TFA, using a microcentrifuge for 30 s at 6000 r.p.m. The beads were washed with 0.5% acetic acid. Peptides were eluted with 40% acetonitrile in 0.5% acetic acid followed by 80% acetonitrile in 0.5% acetic acid. The organic solvent was

removed using a SpeedVac concentrator. The samples were reconstituted in 0.5% acetic acid and stored at −80 °C until analysis.

*Mass spectrometry analysis.* An aliquot of the experimental sample and the control were loaded onto an Acclaim PepMap trap column (2 cm × 75 μm) in line with an EASY-Spray analytical column (50 cm × 75 μm ID PepMap C18, 2 μm bead size) using the auto sampler of an EASY-nLC 1000 HPLC (Thermo Fisher Scientific) with solvent A consisting of 2% acetonitrile in 0.5% acetic acid and solvent B consisting of 80% acetonitrile in 0.5% acetic acid. The peptides were gradient eluted into a Thermo Fisher Scientific Q Exactive mass spectrometer using the following gradient: 5–35% in 60 min, 35–45% in 10 min, followed by 45–100% in 10 min. High-resolution full MS spectra were recorded with a resolution of 70,000 at m/z 400, an AGC target of 1e6, with a maximum ion time of 120 ms, and a scan range from 400 to 1500 m/z. The top 20 MS/MS spectra were collected with an AGC target of 5e4, maximum ion time of 120 ms, one microscan, 2 m/z isolation window, and Normalized Collision Energy (NCE) of 27 and a dynamic exclusion of 30 s.

*Data processing.* The MS/MS spectra were searched against the UniProt *Mus musculus* reference proteome database (downloaded 02/2019) with the flag tagged Pax7 sequence inserted and containing common contaminant proteins using Proteome Discoverer 1.4. The search parameters were as follows: precursor mass tolerance ±10 p.p.m., fragment mass tolerance ±0.02 Da, digestion parameters trypsin allowing two missed cleavages, fixed modification of carbamidomethyl on cysteine, variable modification of oxidation on methionine, deamidation on glutamine and asparagine. The data were filtered using a 1% peptide and protein FDR cut-off searched against a decoy database. Protein–protein interactions were assessed using Significance Analysis of INTeractome (SAINT)[60], in which proteins with spectral fold-change of ≥2 and FDR of their SAINT scores <10% were considered as high-confidence Pax7 interacting proteins.

### Hi-C library preparation for sequencing and pCHi-C.
Two biological replicates of ~30 × 10⁶ iPax7 cells (+/−Dox) were cross-linked for 10 min by 2% formaldehyde in fresh media. Cross-linking was quenched by addition of 0.125 M glycine. Cells were incubated at room temperature for 5 min and then centrifuged at 4 °C for 5 min. Cross-linked cells were washed once in cold PBS, and the cell pellets were flash frozen in liquid nitrogen and stored at −80 °C.

Hi-C library generation was carried out with in-nucleus ligation as described previously[61], with slight modifications. Briefly, chromatin was digested overnight at 37 °C with HindIII (NEB). Digested ends were filled in using biotinylated d-CTP and ligated in preserved nuclei. Cross-links were reversed by proteinase K treatment at 65 °C overnight and DNA was purified by phenol–chloroform extraction. DNA concentration was measured using Qubit 2.0 Fluorometer (Life Technologies). Biotin was removed from non-ligated restriction fragment ends by incubating the Hi-C library DNA with T4 DNA polymerase (NEB) for 4 h at 20 °C in the presence of dATP and dGTP. After DNA purification with phenol: chloroform, 40 μg of DNA was sheared to an average size of 350 bp using Covaris LE220 following the manufacturer's instructions. Double size selection of DNA was performed using AMPure XP beads (Beckman Coulter) and the sonicated DNA was end-repaired with T4 DNA polymerase, T4 DNA polynucleotide kinase, Klenow (all NEB) and dNTPs in 1× T4 DNA ligase reaction buffer (NEB). Biotin-marked ligation products were isolated using MyOne Streptavidin C1 DynaBeads (Invitrogen), adenine-tailed and ligated to paired-end adaptors (Illumina). The immobilized Hi-C products were amplified using either NEBNext Multiplex Oligos for Illumina (NEB) for Hi-C-seq or SureSelect Primers from SureSelectXT Reagent kit, HSQ (Agilent Technologies) for pCHi-C-seq with seven PCR amplification cycles with Phusion High-Fidelity DNA Polymerase (NEB). Paired-end reads for Hi-C libraries were obtained with Illumina Hiseq 2500 (2 × 50 cycles) and NextSeq 500 (75 cycles).

**pCHi-C.** Capture Hi-C of promoters was carried out with SureSelect target enrichment (SureSelectXT Custom 3–5.9 Mb library, Agilent Technologies), using the custom-designed biotinylated RNA bait library that overlap 25,747 Ensembl-annotated promoters of protein-coding, noncoding, antisense, snRNA, miRNA, and snoRNA transcripts[21] and custom paired-end blockers according to the manufacturer's instructions (Agilent Technologies). A total of 500 ng to 1 μg of Hi-C library was used for capture. The enriched library was further amplified for pCHi-C-seq with six PCR amplification cycles using SureSelect primers (SureSelectXT Reagent kit, HSQ, Agilent Technologies) and Phusion High-Fidelity DNA Polymerase (NEB). Paired-end reads for pCHi-C libraries were obtained with Illumina NextSeq 500 (75 cycles).

**Data processing for Hi-C.** All Hi-C datasets were uniformly pre-processed with the HiC-bench platform[62], outlined briefly as follows. First, all paired-end sequencing reads were aligned against the mouse genome version NCBIM37/mm9 with bowtie2 version 2.2.6 (ref. [63]) (specific settings: --very-sensitive-local --local). The aligned reads were further filtered using the GenomicTools[64] gtools-hic filter command with the following parameters: --mapq 30 --min-dist 25000 --max-offset 500 (integrated in HiC-bench), which discards multi-mapped reads ("multihit"),

read-pairs with only one mappable read ("single sided"), duplicated read-pairs ("ds. duplicate"), read-pairs with a low mapping quality of MAPQ <30, read-pairs resulting from self-ligated fragments, and short-range interactions resulting from read-pairs aligning within 25 kb (together called "ds.filtered"). See Supplementary Data 1 for an overview of Hi-C sequencing depth. A Hi-C contact matrix was then generated for each replicate and subjected to iterative correction (ICE; integrated in hic-bench)[65]. TADs were called using the "domains" operation in hic-matrix at 40 kb resolution with default settings[62]. Sub-nuclear compartments were identified using Principal Component Analysis (PCA)[24] using HOMER at 50 kb resolution with a 100-kb window (runHiCpca.pl -res 50000 -window 100000). H2K27ac ChIP-seq peak regions for each sample were used to assess the proper sign of the PC1 results. The Pearson's correlation matrix of each individual chromosome was visualized with Juicebox[66].

**Data processing for pCHi-C**. Raw sequencing reads from pCHi-C were processed using the HiCUP pipeline[67], which maps the positions of di-tags against the mouse genome (NCBIM37/mm9), filters out experimental artifacts, such as circularized reads and re-ligations, and removes all duplicate reads. See Supplementary Data 1 for an overview of pCHi-C sequencing depth. Interaction confidence scores were computed with CHiCAGO[25]. High-confidence interactions were defined as CHiCAGO score ≥5, as described[25]. CHiCAGO scores for interactions from each replicate were used for PCA. Both cis- and trans-interactions were kept, where cis-interactions account for ≥98% of the total number of significant interactions (see Supplementary Data 1 for a detailed quantification). To identify inter-TAD pCHi-C interactions, the TAD boundary regions detected from the Hi-C data were further extended 20 kb up- and down-stream and only high-confidence cis pCHi-C interactions that cross the whole extended boundary regions were considered as inter-TAD interactions. All promoter–promoter interactions and interactions from non-protein-coding promoters were removed before identifying interactions between promoters and active enhancers. To further verify results from the above analyses, we plotted virtual 4C tracks for each bait region, which were normalized according to reads per million (RPM). Genome-wide interactions were visualized using the WashU EpiGenome browser[68]. The P–En networks in Fig. 6b were visualized using Cytoscape[69].

**ChIP-seq and data analysis**. ChIP for Pax7 (Developmental Studies Hybridoma Bank), CTCF (Cat. 07-729, Millipore), and Smc3 (Cat. ab9263, abcam) was performed as described previously[11] with two biological replicates per factor per condition. Briefly, cells were cross-linked at room temperature with 1% formaldehyde for 10 min, and then quenched by incubating in 0.125 M glycine. The cells were washed three times with ice-cold PBS and resuspended in 3 ml of ChIP lysis buffer (10 mM Tris pH 8, 1 mM EDTA, 0.5 mM EGTA, 0.5% N-lauroyl sarcosine, and protease and phosphatase inhibitors) per 100 μl of cell pellet. Each 1 ml cell resuspension was individually sonicated on ice for nine rounds of a 30 s ON/60 s OFF cycle using a Branson Sonifier 450 at Output 3 and constant power to obtain an average fragment size of 200–300 bp. Debris was removed through centrifugation at 20,000×g for 15 min at 4 °C, and the chromatin in the supernatant was quantified according to DNA concentration. For each ChIP reaction, 25 μg was diluted in ChIP lysis buffer plus 1% Triton X-100, 0.1% sodium deoxycholate, 1 mM EDTA, and protease inhibitors before pre-clearing with protein G or A Sepharose (previously blocked in 1 mg/ml BSA) for 4 h at 4 °C. Pre-cleared chromatin was then incubated with 2 μg of antibody overnight at 4 °C. Immunocomplexes were captured by incubating with protein G or A Sepharose for 4 h at 4 °C. Immunoprecipitates were washed eight times with RIPA buffer (50 mM HEPES pH 7.6, 10 mM EDTA, 0.7% sodium deoxycholate, 1% NP40, 0.5 M lithium chloride, and protease inhibitors), once with a low-salt wash (50 mM Tris pH 8, 10 mM EDTA, 50 mM sodium chloride), and eluted in elution buffer (50 mM Tris pH 8, 10 mM EDTA, 1% SDS) at 65 °C for 15 min. The supernatant was incubated overnight at 65 °C to reverse crosslinks, diluted twofold in 50 mM Tris pH 8 plus 10 mM EDTA, and then sequentially digested with 80 μg RNase A for 2 h at 37 °C and 80 μg proteinase K for 30 min at 55 °C. DNA was extracted with phenol/chloroform/isoamyl alcohol and ethanol precipitated. DNA pellets were resuspended in 10 mM Tris pH 8 and quantified using Qubit 2.0 Fluorometer (Life Technologies).

Sequencing libraries were prepared from purified DNA using the NEBNext Ultra™ II DNA Library Prep Kit for Illumina (NEB) following the manufacturer's instruction and reads were obtained with Illumina HiSeq 2500 (single-end for Pax7) and the Illumina NextSeq 500 (75 cycles, paired-end for CTCF and Smc3).

Raw reads from this study and previously published work (Supplementary Table 1) were aligned to the mouse genome version NCBIM37/mm9 with bowtie2 version 2.3.4.1 (ref. [63]) (specific settings for paired-end reads: --local --no-mixed --no-discordant; specific settings for single-end reads: --local). Only uniquely mapped reads were selected for downstream analysis. PCR duplicates were removed using Picard-tools version 1.88. Peak-calling was done using MACS2 version 2.1.1 (ref. [70]) with a default q-value of 0.05, except for Pax7 ChIP-seq with a q-value cut-off of 0.01, Tead1/4 ChIP-seq with a p-value cut-off of 0.1, c-Jun and Foxk1 ChIP-seq with a p-value cut-off of 0.005, and data from GSE36024 with a q-value cut-off of 0.01. Replicate experiments were merged and the data were normalized per million total reads for visualization. Heatmaps and profile plots for normalized ChIP-seq data were generated using deepTools version 3.1.0 (ref. [71]).

**mRNA-seq and Gene Ontology analysis**. mRNA-seq reads from a previous study (Supplementary Table 1) were mapped to the Ensembl annotated genome (NCBIM37/mm9) using STAR version 2.5.0c[72] and differentially expressed genes ($p_{adj} < 0.05$; p value adjusted by FDR) were identified by DESeq2 (ref. [73]) after normalizing each library to number of reads in all protein-encoding genes. Replicate experiments were merged, and the data were normalized per million total reads for visualization. Gene Ontology (GO) term enrichment analysis was performed using DAVID 6.8 (ref. [74]). Similar terms were merged and sorted by the p-value calculated from the Fisher's exact test.

**ATAC-seq and data analysis**. ATAC-seq libraries were prepared as described[75]. Briefly, cells were harvested by trypsinization and resuspended in appropriate media; 500,000 cells were subsequently used for nuclear extraction. Cells were washed with cold PBS and resuspended in cold lysis buffer (10 mM Tris-HCl pH 7.4, 10 mM NaCl, 3 mM MgCl₂, 0.1% IGEPAL CA-630), and centrifuged at 500×g for 10 min at 4 °C. Supernatant was discarded and the nuclear pellet was resuspended in nuclease-free water; 50,000 cells in nuclease-free water was mixed with TD Buffer and 2.5 μl Tn5 Transposase (Illumina cat. #FC-121-130) to a total volume of 50 μl. Transposition occurred at 37 °C for 30 min, after which transposed DNA was purified using a Qiagen MinElute Kit and eluted in 10 μl of Elution Buffer. ATAC libraries were generated by initial amplification for five cycles using the following PCR conditions: 72 °C 5 min; 98 °C 30 s; then cycling at 98 °C 10 s, 63 °C 30 s, and 72 °C 1 min using NEBNext High-Fidelity 2X PCR Master Mix, Forward/Reverse NEBNext index primers and 20 μl of the purified transposed DNA. We used the qPCR plot on LightCycler 480 (Roche), and when needed, used additional cycles of PCR amplification to reach the cycle number corresponding to 1/3 of the maximum fluorescent intensity. Libraries were purified with AMPure (Beckman) beads and the tagmentation was visualized via Tapestation (Agilent).

All libraries were sequenced (2 × 50 cycles, paired end) on an Illumina HiSeq2500 machine. Raw reads were aligned to the NCBIM37/mm9 reference genome with Bowtie2 version 2.2.6 using the options --local --dovetail --minins 38 --maxins 2000 --no-mixed --no-discordant. Only uniquely mapped reads were selected for downstream analysis. PCR duplicates were removed using Picard-tools version 1.88. Replicate experiments were merged, and the data were normalized per million total reads for visualization. Transposase-accessible regions were identified using the callpeak command of MACS2 version 2.1.1 with options --nomodel --nolambda --keep-dup all --call-summits[70]. Replicate experiments were merged and the data were normalized per million total reads for visualization. Heatmaps and profile plots for normalized data were generated using deepTools version 3.1.0 (ref. [71]).

**Calling active enhancers and SEs**. Genome-wide accessible chromatin regions were determined by ATAC-seq and classified using k-means clustering based on binding profiles of histone modifications (H3K4me1, H3K27ac, H3K4me3, and H3K27me3), CTCF and Smc3 as shown in Fig. 2a and Supplementary Fig. 3A, B. Individual active enhancers were defined as non-promoter (>1 kb from the nearest TSS and do not overlap with pCHi-C baits) regions with ATAC-seq, H3K4me1, and H3K27ac ChIP-seq signals. The ATAC-seq peaks called by MACS2 were used to localize open chromatin regions globally and to identify individual active enhancers.

SEs for each cell population were called using the ROSE package with the default stitching size of 12.5 kb[76]. All active enhancers from each cell population were used as input constituent enhancers and input-subtracted H3K27ac ChIP-seq signal from a single donor that showed the highest signal to noise was used for ranking the stitched regions.

**DNA-binding motif analysis**. DNA-binding motif analysis at Pax7 enhancer-containing SEs was done using MEME-ChIP[77,78] with default settings. A list of 1 kb DNA sequences centered on the mid-point of individual active enhancers were used as input for each search.

**CRISPR interference**. To silence enhancer regions with CRISPRi, 20-nt-long sgRNAs against the core of the enhancer (summit of ATAC-seq or Pax7 ChIP-seq signal) were designed using CRISPOR (http://www.crispor.tefor.net). sgRNA against the synthetic CAG promoter was used as a control. sgRNA oligonucleotides were cloned into a self-generated lentiGuide-Zeo vector as previously described[79]. Briefly, oligonucleotides (Thermo Fisher) containing gRNA sequences flanked by BsmBI compatible overhangs were phosphorylated with T4 polynucleotide kinase (NEB) and annealed. Fragments were ligated into BsmBI-digested destination vector. Ligated constructs were transformed into Stellar competent Escherichia coli (Takara) and clones were checked by Sanger sequencing. The lentiGuide-Zeo vector was constructed by replacing the puromycin selectable marker in lentiGuide-Puro (Addgene Plasmid #52963) with a Zeocin selectable marker. Doxtreated iPax7 cells were infected with lenti_dCas9-KRAB-MeCP2 (Addgene Plasmid #122205), selected with blasticidin (10 μg/ml). Medium was replenished every 48 h thereafter, until all negative control cells were dead (usually 4–5 days). After selection, cells were infected with different sgRNAs and selected with Zeocin (1 mg/ml). Medium was replenished every 48 h thereafter, until all negative control cells were dead (usually 14–16 days). The efficiency of the sgRNAs was tested by real-

time PCR. Oligos for sgRNA and real-time PCR are listed in Supplementary Table 2.

### 3D modeling and analysis of chromatin topology

*Data normalization and filtering*. The interaction pCHi-C data were binned at 5 kb resolution, taking into account the size limit of fragments length distribution in the captured regions. Next, the interaction datasets were normalized by 'visibility', that is, proportional to how much each binned locus interacts with the rest of the genome (1);

$$value_{ij} = \frac{cell_{ij}}{\sum row_i + \sum row_j - cell_{ij}}, \qquad (1)$$

where $cell_{ij}$ is raw interaction frequency value between bin $i$ and bin $j$, $row_i$ is row from the whole-genome interaction matrix containing all the interactions involving bin $i$, and $row_j$ is row from the whole-genome interaction matrix containing all the interactions involving bin $j$. Any pair of interacting fragments that included non-pCHi-C baits was considered an artifact and removed from the final interacting matrix.

*Modeled region definition*. The extent of the region to be modeled was defined to contain all key elements of the genomic region (i.e., promoters and enhancers) as well as their genomic 3D context. Specifically, the model included the following features: (i) all interaction bins containing a key element were selected; (ii) any other bin interacting with the key elements (that is, top 3% of all interactions from a key element) was selected; (iii) a network was created between bins where edges correspond to top 3% interactions between any of the selected interactions; (iv) group nodes closer than 25 kb into groups; (v) we removed poorly connected groups (that is, with the ratio between edges and nodes smaller than 5); and (vi) we extracted the connected groups that contain the most key elements.

*3D modeling*. Next, the normalized interaction matrices of selected regions were modeled using TADdyn[80], a molecular dynamic-based protocol implemented in TADbit[81] which is suited for sparse datasets such as pCHi-C. Here we used a similar protocol as previously described[82]. Briefly, the selected chromatin regions were first represented as a bead-spring polymer model with a bead size proportional to the resolution of the experiment. The conformation of this polymer was then initially defined by a random walk and afterward fitted to the inferred spatial restraints by a steered molecular dynamics protocol. The conformation of the resulting model is one of the many possible ones that minimizes the defined scoring function, so to take into account the effect of the population data, a total of 1000 models were generated for each genomic region and dataset. The contact map generated from the ensemble of built models highly correlated with the normalized pCHi-C interaction matrices (Supplementary Table 3). All models for each simulated region were used to further analyze their structural conformations.

*Structural analysis of 3D models*. TADbit was used to calculate several structural measures from the ensemble of models. Those included: (i) particle-to-particle distance distributions represented as box plots obtained from the ensemble of models, and distribution comparisons were obtained by applying a two sample Kolmogorov–Smirnov statistic; (ii) distance distribution from a selected particle of interest to those harboring promoters, enhancers, or CTCF peaks, the distance distributions were displayed in a line plot, centered in the Y axis by the median distance between the focus point and the particle stated in the X axis (error bars correspond to a standard deviation above and below the median); (iii) visual representations of the resulting models were generated using Chimera[83] with the centroid model of the ensemble as a worm-like tube colored with enhancers and promoters in red and blue, respectively.

**Reporting summary**. Further information on research design is available in the Nature Research Reporting Summary linked to this article.

## Data availability

Sequencing data generated from this work are available under the GEO accession numbers 'GSE150638' and 'GSE147057'. Source data from previously published work (also isted in Supplementary Table 1) are available under the following accession numbers: 'GSE56077', 'GSE66901', 'GSE82193', 'GSE56932', 'GSE37525', 'GSE36024', 'GSE29184', 'GSE125203', 'GSE89977', 'GSE35156' and 'GSE95533' from GEO and 'E-MTAB-2414' from the ArrayExpress database. The mass spectrometry data have been deposited to the MassIVE Repository with the dataset identifier 'MSV000086392'. All other relevant data supporting the key findings of this study are available within the article and its Supplementary Information files or from the corresponding author upon reasonable request. A reporting summary for this article is available as a Supplementary Information file. Source data are provided with this paper.

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

## Acknowledgements

We are most grateful to S. Schoenfelder and P. Fraser for their generous assistance in setting up pCHi-C in our laboratory, and M. Schober and J. Skok and A. Califano for advice and critical reagents. This work was supported by funding to B.D.D. from the NIH (1R21AR068786-01A1 and 1R01GM122395) and funding to R.C.R.P. from the NIH (1R01AR055299-01A1 and 1R01AR071439-01). We thank the NYU School of Medicine Proteomics and Genome Technology Cores for assistance. The mass spectrometric work is supported in part by NYU Grossman School of Medicine and the Laura and Isaac Perlmutter Cancer Center Support grant P30CA016087 from the National Cancer Institute. We also thank D. Darling for his assistance with graphics. B.D.D. would also like to thank the NYU Center for Skeletal and Craniofacial Biology for generously providing a pilot grant. This research was partially funded by the European Union's H2020 Frame-work Programme through the ERC (grant agreement 609989 to M.A.M-R.). We also acknowledge the support of Spanish Ministerio de Ciencia, Innovación y Universidades through BFU2017-85926-P to M.A.M-R. CRG thanks the support of the Spanish Ministry of Science and Innovation to the EMBL partnership, the 'Centro de Excelencia Severo Ochoa 2013-2017', SEV-2012-0208, the CERCA Programme/Generalitat de Catalunya, Spanish Ministry of Science and Innovation through the Instituto de Salud Carlos III, the Generalitat de Catalunya through Departament de Salut and Departament d'Empresa i Coneixement and the Co-financing by the Spanish Ministry of Science and Innovation with funds from the European Regional Development Fund (ERDF) corresponding to the 2014-2020 Smart Growth Operating Program.

## Author contributions

Methodology: N.Z., J.M-E.; investigation: N.Z., J.M-E.; writing—original draft preparation: N.Z., B.D.D., J.M-E., M.A.M-R.; writing—reviewing and editing: N.Z., A.M., K.C.L., R.C.R.P., A.T., J.M-E., M.A.M-R., B.D.D.; supervision of computational methods: A.T., M.A.M-R.; overall supervision of experimental procedures and research plan: B.D.D. Funding acquisition: R.C.R.P., M.A.M-R., B.D.D.

## Competing interests

The authors declare no competing interests.
