## [Peer Review File · Nature Communications]

Reviewers' comments:

Reviewer #1 (Remarks to the Author):

The manuscript by Zhang et al., entitled "Changes in chromatin topology drive muscle progenitor specification and myogenic differentiation" employs a mouse embryonic stem cell (ESC) system that enables doxycycline inducible expression of Pax7. Upon inducible expression of Pax7 these cells take on the properties of myogenic progenitors; even after the removal of doxycycline indicative of stable reprogramming. Due to the scarcity and difficulty in culturing adult muscle stem cells, the authors propose using the Pax7 inducible ESC system (iPax7-ESC) to gain insights into the organization of promoter and enhancer interactions that confer myogenic potential. The authors reason such interactions could be observed in adult muscle stem cells. The manuscript employs an impressive array of methodologies and bioinformatics. Based on their observations the authors describe many potential promoter enhancer interactions, and enhancer hubs that could confer myogenic potential in their iPax7-ESC cells and possibly adult muscle stem cells.

The work presented in this manuscript follows a previous publication by the same group (i.e.: Lilja et al., 2017), that demonstrate that induced pluripotent stem cells (iPSCs) derived from skeletal muscles precursor cells where the sustained induced expression of Pax7 activate the myogenic lineage program. They found that Pax7 binds in open and accessible chromatin regions away from genes' TSS and is excluded from H3K27me3 enriched loci. While Pax7 render MyoD sites accessible, the work suggested that Pax7 may play a more pronounced role at enhancers (recruitment of co-factors and epigenetic modifiers). Also, Pax7 is unable to remodel facultative heterochromatin. It is not clear why they changed their system to mESC, as some of the conclusion in this manuscript do not always support previous findings, and they do not discuss those points.

Despite the variety of experiments employed, the results are described in a very superficial manner; thus, it is difficult to comprehend the various conclusions made throughout the manuscript. Although this reviewer appreciates the difficulties in conducting the proposed methods from sorted adult muscle stem cells, no effort is made to demonstrate that any of the possible promoter enhancer interactions occur in adult muscle stem cells. Therefore, the relevance and advance this work provides to the study of adult muscle stem cell biology and regenerative medicine is unclear. Overall the manuscript is descriptive without needed follow up of identified promoter enhancer, or enhancer super hubs.

Some points:

Figure 2A: Provide Pax7 binding profile

Figure 2F: Provide Pax7 binding

- It isn't clear what is the direct contribution of Pax7 as opposed to MyoD. You previously showed that iPax7 enhance MyoD binding sites accessibility (Lilja et al., 2017). Here, you show that MyoD promoter itself is more accessible and interact more with distal regulatory elements. MyoD can also binds its own promoter to regulate its expression and is a master regulator of myogenesis.

A recent publication (Dall'Agnese et al., 2019) showed that MyoD can substantially reshape the chromatin structure and re-wire chromatin interactions between cis-regulatory elements (especially promoter-enhancer interactions), and by doing so promote myogenesis. These data in many ways match what you found in iPax7 cells. Yet, you show that MyoD KO decrease chromatin accessibility independently from Pax7 and that induction of iPax7 does not alter MyoD binding (Sup 5/Sup7). MyoD alone can reprogram a fibroblast into a myoblast. So:

- 1) How do you reconcile these findings from your past studies and studies from others with this manuscript?

2) Do you expect Pax7 to act entirely independently from MyoD activity in progenitors, despite the fact that Pax7 seems to facilitate MyoD binding?

3) Pax7 and MyoD often overlap during myogenesis and even share some target genes. It is hard to imagine that Pax7-mediated chromatin re-wiring would not influence MyoD-mediated re-wiring of P-En interactions.

4) Deletion/Knockdown of MyoD in iPax7 cells after Dox induction would help decipher the direct role of Pax7.

- During mouse muscle development, Pax3 was shown to act as a pioneer transcription factor more so than Pax7 (Especially upstream of the Six/Pitx genetic cascades). Did you assess Pax3 status in your cells? It is important considering that Pax3 can activate similar cascade than Pax7 (Notch/HeyL, MyoD, Six/Eya...) and reprogram naïve cells into myoblast (Sato et al., 2019).

- How do your data relate to genome-wide data obtained from actual muscle stem cells? (ATAC-seq, Hi-C etc.). While the mechanistic work is impressive, it would be of interest to confirm some of the data in actual muscle stem cells (satellite cells) since iPax7 remains an artificial system.

- The authors claim to have identified a Pax7-bound enhancer hub that regulates the essential myosin heavy chain cluster. This is entirely based on bioinformatics, and there are no experiments or data to substantiate the conclusion. For example, some sort of assay that demonstrates the identified region has enhancer properties with regards to myosin heavy chain gene regulation.

Reviewer #2 (Remarks to the Author):

The authors provide a comprehensive epigenomic ((C)-HiC, ATAC-seq, ChIP-seq, transcriptome) profile of a previously validated model for myogenic differentiation, focussed on the characterization of promoter-enhancer interactions, particularly comparing those which are maintained or specific when comparing muscle precursors to myotubes (+/- Dox, respectively in them model). Their major conclusions are novel, particularly with respect to the myogenesis field, but are perhaps not completely surprising. In particular:

- * Pax7 (a previously identified pioneer factor for this developmental point) is enriched at clusters and higher-order networks of promoter-enhancer interactions, one such hub (at the Myh cluster) being validated by CRISPRi.

- * Interactions that are maintained on the loss of Pax7 (Dox- condition) involve regions bound by a plethora of other muscle-specific TFs.

I support publication in Nature Communications, subject to some minor modifications:

1. The work is, for the most part, correlative. This is not a problem in itself, since the authors have already commented on the redundancy of the non-Pax7 TFs in maintaining the "conserved" interactions, and elegant experiments altering chromatin topology without altering other aspects of chromatin are extremely hard to perform adequately. But the authors should downplay their hints at causation (in particular, words like "deterministic" which crop up in the text, and the title, saying that "changes in chromatin topology DRIVE ... myogenic differentiation").

2. The genome-wide analyses are sound and convincing, but the specific examples are less so. There are many problems with Circos plots on the WashU browser which reduce their usefulness. They either show all interactions (which makes a confusing mess) or only those interactions that are considered "significant" (i.e. Chicago score ≥ 5), which removes a lot of the information, such as whether the interaction is a random "spike" or is also supported by smaller interactions in the flanking regions,

creating a believable "peak". Further, the colour-coding for the "strength" of the interaction (either sequence count or Chicago score) is not very dynamic, and it is difficult to spot differences beyond presence/absence (which again, is artificial if a cutoff threshold is used). I recommend the authors explore different other possibilities to plot bait-specific interactions as virtual 4C plots or the like to show true and quantitative differences in their called interactions. Unfortunately, in my experience a large number of interactions that appear convincing in the statistics and meta-analyses can look quite disappointing when REALLY plotted.

3. It is unclear how the clusters in figures 2A and figures S3A and B are derived. The information is missing in the materials and methods.

Reviewer #3 (Remarks to the Author):

Zhang et al present a manuscript in which they analyze distal enhancer gene regulation in a muscle differentiation model. The previously established model is an interesting model to study gene regulation in muscle differentiation. The authors show that there are genes that interact with putative distal enhancers. There is a plethora of modest enrichments that are suggestive of regulatory regions that play a role in muscle differentiation. The authors show an interaction between the transcription factor Pax7 and the architectural protein CTCF. Unfortunately, no deep new insights are gained from these analyses. The CRISPRi experiment on the enhancer in the final figure is interesting, but hardly surprising.

My main criticism is that the interesting parts of the paper are difficult to tease out because of the high information density in all the figures. Most of which is difficult parse for the average reader. The authors should focus on the important information. Examples: Figure 2A: location of the peaks in the genome there are 11(!) different categories, the result is not very interesting or important; Supp. Figure 7D there is "modest" effect, the authors should consider that the effect size is very small, why is this relevant.

Hi-C analyses have shown that the vast majority of loops are CTCF mediated. Recent work from the Mundlos lab (Despang et al.) has shown however, that most CTCF sites are irrelevant for the regulation of genes. These analyses were done with high-resolution capture Hi-C of an entire locus. Here, the authors use HindIII as a restriction enzyme for the (pC)Hi-C. This has a relatively low resolution. This means many of the interactions that will be picked up are actually CTCF-CTCF loops that are close to a promoter. I find the conclusion made by the authors that CTCF functions or may function in a spatial interaction network that facilitates muscle-specific gene regulation too strong. This is based mostly on low resolution correlations and functional validation is needed if the author want to make this point.

To show that CTCF is important for the regulation the genes in question the authors should mutate the CTCF binding sites in the vicinity of the interacting enhancers that they find to see if this affects the expression of the regulated genes.

The authors should also refrain from making bombastic claims about their dataset, it does not add to the information value of the manuscript. The authors state "3D modeling confirmed" however, the 3D modeling is a result of the data that is fed into the modeling process, it can therefore hardly be seen as confirmation of the data. The manuscript is filled with conclusions that are highly speculative. For instance: "These findings confirm that we have identified interactions unique to muscle progenitors that direct expression of master muscle regulatory factors (also see below)." An interaction is found and gene expression is found. However, it is unclear whether this interaction directs gene expression. The manuscript is riddled with sloppy reasoning like this. Furthermore, there is very little room for alternative explanations.

One major claim is that Pax7 interacts with CTCF. However, the mass spec data is not very convincing to me. There are a number of issues:

- (1) What is the control in this experiment? Without a proper control the authors cannot make the claim that this interaction is specific. Especially because the scores are quite low.
- (2) The authors should visualize the MS data in a volcano plot. Related to this: how many replicates did the authors perform?
- (3) The number of peptides is not the most important information. The authors should show the enrichment between Pax7-flag / control-flag with the pull-down and label-free approach. They need to perform at least 3 replicates.

Reviewer 1

We would like to respectfully emphasize that while MyoD is clearly a pivotal muscle regulatory factor, we have chosen to focus on Pax7, which acts prior to MyoD. MyoD acts at a later point in differentiation, and it does not play a role in myogenic precursor specification, which is the focus of this paper. Indeed, MyoD is not expressed in quiescent satellite cells. Further, Reviewers previously commented on the large volume of data presented in this paper, and we believe that introducing new studies on MyoD would increase the amount of data further, would serve to diffuse the focus of our studies, and would constitute a completely independent line of investigation beyond the scope of the current study. It is also important to point out that the Reviewer cites data from the myogenic conversion studies (Dell'Agnese). These studies are very informative, although they are artificial in this context because MyoD is never expressed in cells (fibroblasts) that have also never expressed Pax7 (or Pax3) before.

Specific points (our response is in **bold**):

1. It is not clear why they changed their system to mESC, as some of the conclusion in this manuscript do not always support previous findings, and they do not discuss those points. **It is possible that the Reviewer may have overlooked certain technical details and conclusions in our current and previous papers. Our current iPax7 system was derived from mESC, and all findings from our previous study were derived from the same cell model. We did not modify our system, and we do not state any conclusions that are inconsistent with previous findings.**

2. Although this reviewer appreciates the difficulties in conducting the proposed methods from sorted adult muscle stem cells, no effort is made to demonstrate that any of the possible promoter enhancer interactions occur in adult muscle stem cells.

We thank the Reviewer for this suggestion, and in an ideal situation, we would pursue this line of investigation. However, it is currently not feasible to perform the suggested experiments for the following reasons. The experiment requires a large number of cells, and even if we had the mice, it would not be possible to obtain enough cells to do the proposed experiments (much less perform the experiment in duplicate or triplicate), since endogenous satellite cells are present at very low abundance. The satellite cells would also need to be processed immediately after FACS sorting to prevent spontaneous differentiation. Therefore, while we agree that this would be a worthwhile line of experimentation if technically feasible, it is important to emphasize previous findings with iPax7 cells, which have documented their ability to emulate authentic myogenic progenitor cells that are able to repopulate the stem cell niche when injected into mouse tissue (Darabi et al., 2011a, 2011b; Incitti et al., 2019). Injection of these cells ameliorates the defects of muscular dystrophy, confirming that they are indeed an authentic source of progenitors. This system, we believe, therefore accurately emulates the satellite cells resident in natural muscle tissue.

3. Overall the manuscript is descriptive without needed follow up of identified promoter enhancer, or enhancer super hubs.

Respectfully, we must firmly disagree with this statement. We believe that our manuscript is based on solid experimental data, and our observations have been validated using orthologous experimental approaches, including proteomics and epigenome editing. Nevertheless, we have now buttressed our findings with additional functional evidence to support our identification of regulatory hubs and cliques. To this end, we have now functionally validated two types of complex topological motifs using epigenome editing. First, in Fig. 7C, we used CRSPRI to reveal an important function for a novel enhancer hub within the *Myh* cluster. Secondly, we have now performed additional experimentation on the *Dmrt2* gene to validate an enhancer clique (newly added Fig. 4H). Thus, we have provided conclusive functional relevance for an enhancer clique and a hub in regulating myogenic gene expression. Third, we have now performed additional proteomic experiments and have merged this analysis with ChIP-seq and pChIC studies to reveal novel feed-forward loops in which Pax7 physically interacts with other myogenic factors to drive expression of muscle stem cell identity and differentiation genes. We believe that, in total, the above functional studies provide robust follow-up support for our genome-wide analyses.

Some points:

Figure 2A: Provide Pax7 binding profile

We thank the Reviewer for the suggestion. We have added the Pax7 binding profile to this figure.

Figure 2F: Provide Pax7 binding

We have added this information to the figure.

- It isn't clear what is the direct contribution of Pax7 as opposed to MyoD. You previously showed that iPax7 enhance MyoD binding sites accessibility (Lilja et al., 2017). Here, you show that MyoD promoter itself is more accessible and interact more with distal regulatory elements. MyoD can also binds its own promoter to regulate its expression and is a master regulator of myogenesis.

With respect to this criticism, we had not shown that iPax7 (we assume that the Reviewer meant Pax7) enhances MyoD sites accessibility in our previous study (Lilja et al., 2017), nor do we draw this conclusion in the present manuscript. During adult skeletal muscle differentiation, *Pax7* expression precedes that of *Myod1*, and our data also suggest that *Myod1* is a Pax7 target, which in no way contradicts the finding that MyoD is able to bind to its own promoter. Because Pax7 is our focus, we did not investigate MyoD binding in this paper, although we agree that MyoD is a master regulator of myogenesis and that it can bind its own promoter. Please see additional comments below.

A recent publication (Dall'Agnese et al., 2019) showed that MyoD can substantially reshape the chromatin structure and re-wire chromatin interactions between cis-regulatory elements (especially promoter-enhancer interactions), and by doing so promote myogenesis. These data in many ways match what you found in iPax7 cells. Yet, you show that MyoD KO decrease

chromatin accessibility independently from Pax7 and that induction of iPax7 does not alter MyoD binding (Sup 5/Sup7).

We believe that this Reviewer may have been confused by our presentation. First, we did not show, nor did we claim, that MyoD loss decreases chromatin accessibility independently from Pax7. The MyoD KO data (original Supplemental Fig. 7D) suggested that MyoD can bind to the same Pax7-associated enhancers that form stable loops with their cognate promoters during cell differentiation and can subtly regulate accessibility. Clearly, however, it is not the only factor that regulates accessibility at these sites. This was our primary reason for showing these data. In response to these comments (and those of Reviewer 3), and to avoid potential confusion, we have removed the original Supplemental Fig. 7D from the revised manuscript, although we are happy to restore it at the discretion of the Editor and/or Reviewers.

Second, our data does not show that ‘induction of iPax7 does not alter MyoD binding,’ and we did not intend to imply that is the case. Pax7 and MyoD are expressed sequentially during muscle cell differentiation, with a small window of overlap. Simply put, we assert that MyoD can bind to the same Pax7-associated enhancers that form stable loops with their target promoters (new Fig. 4B). Therefore, it is possible that Pax7 binding can influence or pre-determine MyoD binding to certain regions, but it is unlikely to directly alter MyoD binding *per se*. Understanding such mechanisms would entail a whole new area of investigation, and therefore, we have not investigated it further in our current study. Again, we believe that removing Supplemental Fig. 7D from the revised manuscript may bypass any potential confusion. The heatmap in original Sup Fig. 5 (new Sup Fig. 6C) shows clustering for all enhancers with interactions, and we have shown the clustering of all Pax7 enhancers that loop to promoters in new Fig. 4B.

MyoD alone can reprogram a fibroblast into a myoblast. So:

1) How do you reconcile these findings from your past studies and studies from others with this manuscript?

Please see above responses. Our studies describe how Pax7 mechanistically regulates establishment of myogenic progenitors, a system unrelated to reprogramming of fibroblasts. Indeed, we believe that comparing fibroblast reprogramming with myogenic progenitor establishment or differentiation from progenitors is not straightforward because fibroblasts never naturally express either MyoD or Pax7. We are not sure what the reviewer means by “reconciling our findings from past studies and studies from others.” We do, however, wish to emphasize that there are no inconsistencies between our current and previous work or work from other laboratories.

2) Do you expect Pax7 to act entirely independently from MyoD activity in progenitors, despite the fact that Pax7 seems to facilitate MyoD binding?

Please see our comments above. Respectfully, the focus of our paper is not MyoD, and our intent was not to examine dependencies of binding. This would entail a completely separate line of investigation. More importantly, it is essential to point out that Pax7 acts upstream of MyoD, and in fact, MyoD is not expressed in quiescent satellite cells when Pax7 is expressed

(although there is a window of time when Pax7 is expressed with MyoD after satellite cell activation). Thus, we believe that Pax7 function can be regarded as largely independent from MyoD in adult muscle progenitors. In addition, there may be multiple, intervening events that occur between the time that Pax7 binds and MyoD is recruited later in differentiation, and this would be worthy of further study in the future.

3) Pax7 and MyoD often overlap during myogenesis and even share some target genes. It is hard to imagine that Pax7-mediated chromatin re-wiring would not influence MyoD-mediated re-wiring of P-En interactions.

Please see our response to Point #2 above. The overlap in binding sites is described in our study (please see new Fig. 4B), and we agree with the Reviewer that there is likely to be cross-talks between these two factors. However, as stated above, the focus of our paper is Pax7, not MyoD, and pursuing this line of enquiry would represent an entirely new area outside the scope of the current manuscript. We agree with the Reviewer that this is a very interesting suggestion, and this will be explored in the future.

4) Deletion/Knockdown of MyoD in iPax7 cells after Dox induction would help decipher the direct role of Pax7.

As detailed above, it is known that Pax7 functions upstream of MyoD, and our system depends entirely on Pax7 expression. Therefore, what we observe in this system should largely reflect a direct impact of Pax7. The impact of MyoD expression may or may not relate to prior Pax7 expression, so we respectfully suggest that the proposed knock-down experiment would be a less direct (or even indirect) measure of Pax7 activity. Moreover, importantly, the impact of MyoD on gene expression has been well-documented by a number of groups previously, as the Reviewer has pointed out. Lastly, as Reviewer 2 correctly stated, “elegant experiments altering chromatin topology without altering other aspects of chromatin are extremely hard to perform adequately.” Depleting MyoD would likely alter chromatin topology based the prior study cited (Dall’Agnese et al., 2019).

- During mouse muscle development, Pax3 was shown to act as a pioneer transcription factor more so than Pax7 (Especially upstream of the Six/Pitx genetic cascades). Did you assess Pax3 status in your cells?

Pax3 and Pax7 are expressed at different developmental stages. Indeed, in response to this question, we analyzed Pax3 expression in our RNA-seq experiment and found that it is not expressed in our iPax7 cells.

- How do your data relate to genome-wide data obtained from actual muscle stem cells? (ATAC-seq, Hi-C etc.). While the mechanistic work is impressive, it would be of interest to confirm some of the data in actual muscle stem cells (satellite cells) since iPax7 remains an artificial system.

Please see Point #2 above. We have obtained ATAC-seq data for satellite cells and have already compared these data with our iPax7 cells (Lilja et al., 2017). As shown in Lilja et al., these data from satellite cells strongly comport with data from iPax7 cells, further attesting to the robustness and authenticity of our iPax7 model. However, Hi-C and pChi-C data

are not available for satellite cells. For the reasons detailed in Point #2 above, this experiment is not currently feasible given the number of cells required. Nevertheless, we respectfully but firmly disagree with the notion that our iPax7 system is artificial: as the foundation of this system, injection of these myogenic precursors can reconstitute the satellite cell niche *in vivo* and ameliorate symptoms of muscular dystrophy in a mouse model. The utility of this model is also well-supported by a number of publications (Darabi et al., 2008a; Darabi et al., 2008b; Darabi et al., 2011a; Darabi et al., 2011b; and Darabi et al., 2012; Carrió et al., 2016; Incitti et al., 2019).

- The authors claim to have identified a Pax7-bound enhancer hub that regulates the essential myosin heavy chain cluster. This is entirely based on bioinformatics, and there are no experiments or data to substantiate the conclusion. For example, some sort of assay that demonstrates the identified region has enhancer properties with regards to myosin heavy chain gene regulation.

We surmise that the Reviewer may have overlooked data presented in our original manuscript. As stated above, we have performed a CRISPRi experiment (Fig 7c) that indeed confirms, *in vivo*, the functional importance of the enhancer hub identified in the *Myh* cluster. In addition, to further address this Reviewer's concerns, we now present additional data on the *Dmrt2* locus (new Fig. 4H), which further supports our pChIC experiments and analyses. Thus, importantly, we now provide functional experimental evidence for both an enhancer hub and a P-En clique.

We thank the Reviewer for his/her comments.

Reviewer 2

I support publication in Nature Communications, subject to some minor modifications.

We are grateful that the Reviewer recommended publication with some minor modifications. We thank the Reviewer for a number of valuable suggestions and for prompting us to perform the requested analyses, detailed below, as they have solidified the overall conclusions in the manuscript. This Reviewer was primarily concerned about our analysis of pChIC data solely through use of Chicago, and we have now conclusively addressed this concern with additional analyses.

1. The work is, for the most part, correlative. This is not a problem in itself, since the authors have already commented on the redundancy of the non-Pax7 TFs in maintaining the "conserved" interactions, and elegant experiments altering chromatin topology without altering other aspects of chromatin are extremely hard to perform adequately. But the authors should downplay their hints at causation (in particular, words like "deterministic" which crop up in the text, and the title, saying that "changes in chromatin topology DRIVE ... myogenic differentiation").

We agree with the Reviewer that experiments to alter chromatin topology without changing other aspects of chromatin are indeed extremely challenging to perform adequately. We also agree with the Reviewer's suggestions regarding causation and have therefore removed all

such claims from the manuscript.

2. The genome-wide analyses are sound and convincing, but the specific examples are less so. There are many problems with Circos plots on the WashU browser which reduce their usefulness. They either show all interactions (which makes a confusing mess) or only those interactions that are considered "significant" (i.e. Chicago score ≥ 5), which removes a lot of the information, such as whether the interaction is a random "spike" or is also supported by smaller interactions in the flanking regions, creating a believable "peak". Further, the colour-coding for the "strength" of the interaction (either sequence count or Chicago score) is not very dynamic, and it is difficult to spot differences beyond presence/absence (which again, is artificial if a cutoff threshold is used). I recommend the authors explore different other possibilities to plot bait-specific interactions as virtual 4C plots or the like to show true and quantitative differences in their called interactions.

Unfortunately, in my experience a large number of interactions that appear convincing in the statistics and meta-analyses can look quite disappointing when REALLY plotted.

This is a valuable suggestion, and we are grateful to the Reviewer for prompting us to add these information. As suggested, we have now generated virtual 4C plots to show true, quantitative differences and have now added these data to the manuscript. Please see new Figure panels (Fig. 2E, Fig. 3J, Fig. 4E, Fig. 7A, SupFig. 4, SupFig. 5A and SupFig. 7C), which substantiate the robustness of our methods and prior analyses. Therefore, these new plots reinforce our conclusions and strengthen the manuscript, and we have modified the discussion accordingly.

3. It is unclear how the clusters in figures 2A and figures S3A and B are derived. The information is missing in the materials and methods.

We apologize for the oversight. We had originally stated this information in the figure legend but have now also added this information to the Materials and Methods section.

We are grateful to this Reviewer for his/her thoughtful review, which has allowed us to significantly improve our manuscript.

Reviewer 3

My main criticism is that the interesting parts of the paper are difficult to tease out because of the high information density in all the figures. Most of which is difficult parse for the average reader. The authors should focus on the important information. Examples: Figure 2A: location of the peaks in the genome there are 11(!) different categories, the result is not very interesting or important

We agree. We thank the Reviewer for these suggestions and apologize that our presentation was, at times, somewhat dense. Genome-wide analyses such as ours aim for comprehensiveness, but by their very nature, they contain a considerable amount information. This is especially true because our study is the first report of Pax7-associated 3D changes in chromatin in muscle progenitors, and as such, we included additional supplemental data to support our system. Admittedly, we still find it challenging to strike the correct balance between too little and too much information. In response to the Reviewer's

concerns, we have attempted to streamline our presentation without resorting to ‘data not shown,’ which is not acceptable to most journals or reviewers. To this end, we have removed or modified/simplified a total of 19 figure panels and a table. Specifically, we have removed 11 figure panels (original SupFig. 1C, D, H, SupFig. 2B, SupFig. 3G, SupFig. 7B-E and SupFig. 9D, E), simplified 7 figure panels (original Fig. 2A, Fig. 5B, SupFig. 1I, J, SupFig. 3A, B, and SupFig. 9F), and modified original SupFig. 7A (new SupFig. 7E) and SupTable 5. In addition, we have moved Figs. 4A, 6C, and 6E to the supplement. We agree that this has reduced the density and significantly improved the presentation of our work. We thank the Reviewer for suggesting ways to streamline our manuscript.

We have also expanded our analysis of one point that may have been overlooked in the previous version of our manuscript owing to the density of our figures, namely, the analysis of feed-forward loops in our myogenic progenitors. This analysis was predicated on our highly reproducible proteomic studies, further detailed below. Please see new Fig. 5A, C. We hope that the Reviewer will agree that this analysis (1) further reinforces the novelty of our work using myogenic precursors and (2) integrates and emphasizes multiple aspects of our work, making the study more cohesive.

Supp. Figure 7D there is “modest” effect, the authors should consider that the effect size is very small, why is this relevant

These effects are statistically significant based on our statistical analyses presented in the original manuscript. However, based on this somewhat modest impact and because it may have resulted in confusion on the part of Reviewers, we have decided to remove this figure panel altogether. Removing these data does not, in our opinion, result in loss of important information, but we are certainly willing to restore it at the discretion of the Editor and the Reviewer.

Hi-C analyses have shown that the vast majority of loops are CTCF mediated. Recent work from the Mundlos lab (Despang et al.) has shown however, that most CTCF sites are irrelevant for the regulation of genes. These analyses were done with high-resolution capture Hi-C of an entire locus. Here, the authors use HindIII as a restriction enzyme for the (p)Hi-C. This has a relatively low resolution. This means many of the interactions that will be picked up are actually CTCF-CTCF loops that are close to a promoter. I find the conclusion made by the authors that CTCF functions or may function in a spatial interaction network that facilitates muscle-specific gene regulation too strong. This is based mostly on low resolution correlations and functional validation is needed if the author want to make this point.

To show that CTCF is important for the regulation the genes in question the authors should mutate the CTCF binding sites in the vicinity of the interacting enhancers that they find to see if this affects the expression of the regulated genes.

We thank the Reviewer for his/her important suggestions, and we apologize for stating our conclusion too strongly or in a confusing manner. To clarify, the result mentioned by the Reviewer for HiC pertained to co-localization of CTCF at TAD boundaries. We did not show the co-localization of CTCF or cohesin with enhancer loops, except for Pax7-associated loops (original Sup. Fig. 7A). To examine this more closely, we have now re-analyzed the occurrence

of CTCF sites at Pax7 enhancer-promoter loops and have added a new figure panel (new SupFig. 7E) to the manuscript to address this point. The Reviewer is indeed correct, and as a result of this new analysis, we have modified our conclusions as per the suggestions above (p. 12). Our results are consistent with what other groups have observed, namely, that CTCF/cohesin co-localize extensively at TAD boundaries, but most of the Pax7 enhancer-promoter loops are not directly formed through CTCF/cohesin.

We respectfully suggest that interpretation of the proposed gene-editing experiment could be confounded by a number of important factors. First, our data and the new analyses described above show that CTCF preferentially binds to the promoters (rather than enhancers) in the Pax7-associated loops (please see new Sup. Fig. 7E), and inhibition of promoters through gene-editing could indirectly alter promoter function (by introducing novel binding events or occluding binding of factors to the promoter), irrespective of enhancer looping. More importantly, because CTCF is only one of several factors that contributes to looping in muscle precursors, it is not clear that removal of its binding sites will have an impact. Indeed, based on our new analysis in new Sup. Fig. 7E, ablating CTCF sites is unlikely to have more than a limited impact, and it may be challenging to choose, *a priori*, those sites that could be impacted by CTCF binding site ablation. It may also be necessary to choose many sites to reach a general conclusion regarding CTCF activity. Further, the proposed CRISPR experiment would likely take several months to complete and validate, and we do not believe that it will alter the conclusions that we have drawn in our revised manuscript. Nevertheless, we believe the Reviewer's comments are noteworthy and have discussed the role of CTCF--and questions that remain--in the Results and Discussion sections of the revised manuscript.

The authors should also refrain from making bombastic claims about their dataset, it does not add to the information value of the manuscript. The authors state "3D modeling confirmed" however, the 3D modeling is a result of the data that is fed into the modeling process, it can therefore hardly be seen as confirmation of the data. The manuscript is filled with conclusions that are highly speculative. For instance: "These findings confirm that we have identified interactions unique to muscle progenitors that direct expression of master muscle regulatory factors (also see below)." An interaction is found and gene expression is found..... Furthermore, there is very little room for alternative explanations.

We thank the Reviewer for this feedback and have rephrased all conclusions throughout the manuscript that may be regarded as speculative or bombastic.

One major claim is that Pax7 interacts with CTCF. However, the mass spec data is not very convincing to me. There are a number of issues:

(1) What is the control in this experiment? Without a proper control the authors cannot make the claim that this interaction is specific. Especially because the scores are quite low.

We apologize for any lack of clarity in our description of this experiment. These cells were generated as described for iPax7 cells except that, instead of Pax7, a single copy of Flag-tagged Pax7 was inserted next to the *Hprt* locus, and expression of Pax7 in these cells is Dox-inducible. The control is the -Dox condition, as stated in the manuscript. Importantly, we have also validated our proteomic experiment with immunoprecipitations and western

blotting (new Fig. 5B), as we believe that this is a standard method to confirm proteomic studies. We have also shown that our results with CTCF (and the other factors presented herein) are reproducible in multiple experiments: we have performed the proteomic experiment with three independent biological samples and have similarly performed the immunoprecipitations and western blotting with three biologically independent replicates. Thus, we have obtained identical results with CTCF and several other factors no fewer than six times with independent samples.

(2) The authors should visualize the MS data in a volcano plot. Related to this: how many replicates did the authors perform?

We thank the Reviewer for these valuable suggestions. We have generated MS data from three independent biological replicates, repeating our prior proteomic analysis presented in the original manuscript to generate a more robust statistical analysis. We have added these data to the revised manuscript, and we have performed the requested analyses using volcano plots (new Fig. 5A). As shown in these plots, enrichment of CTCF and the highlighted factors is highly significant based on fold-enrichment and Significance Analysis of INTERactome (SAINT) (Choi et al. 2011) scores. We thank the Reviewer for prompting us to perform experiments and analyses that have vastly strengthened our conclusions (please also see below).

(3) The number of peptides is not the most important information. The authors should show the enrichment between Pax7-flag / control-flag with the pull-down and label-free approach. They need to perform at least 3 replicates.

As stated in Point #2 above, we have now provided the requested data as suggested (new SupTable 5). The data are based on three replicates. As suggested, we have also plotted spectral fold-enrichment and have convincingly shown that the proteins of interest are indeed significantly enriched in a volcano plot.

We thank the Reviewer for prompting us to more rigorously analyze our proteomic data. Indeed, this effort allowed us to solidify a novel aspect of our work, mentioned above, that may have been overlooked previously owing to the volume of data we presented. Importantly, our proteomic analysis identified a cohort of Pax7-associated, sequence-specific transcription factors (TFs) that were notable for the following reasons. First, many (23 out of 68) of the genes encoding these Pax7-associated TFs were also identified as targets of Pax7 enhancers in our pChIC experiments. Second, many of these factors localized near Pax7 binding sites in enhancers that loop to promoters by CHIP-seq in muscle (e.g., Jun, Runx1, Tead1, and others; please see revised Fig. 4B and 5A). Third, several of these Pax7-associated TFs were shown to play an essential role in muscle stem cell function (*Eya4*, *Six1*, *Runx1*). Forth, by using a *de novo* motif discovery algorithm (MEME) and exploring CHIP-seq binding profiles for several TFs, we found the Pax7-bound super-enhancers in iPax7 cells were enriched for motifs and binding sites of many of these factors (revised Fig. 5F, G). Together, these findings suggest a model in which these TFs play roles in feed-forward regulatory loops through collaborative, physical interactions with Pax7 at enhancers and super-enhancers. To emphasize this point, we have illustrated this model in new Fig. 5C. We hope that the Reviewer agrees that this analysis further highlights the novelty of our work in myogenic

precursors.

We thank this Reviewer for his/her thoughtful and rigorous review which has allowed us to significantly improve the presentation and clarity of our manuscript.

Reviewer #1 (Remarks to the Author):

All concerns addressed

Reviewer #2 (Remarks to the Author):

The authors have replied satisfactorily to my comments and made suitable changes. I therefore support the manuscript's publication in Nature Communications.

Reviewer #3 (Remarks to the Author):

I feel the authors have addressed most of my comments (i.e. more replicates for IP-MS, which is crucial). They have decided to tone down their conclusions with regard to the role of CTCF in Pax7-associated loops.

However, I would like to note that the IP-MS was not controlled as previously suggested. The control is a cell line without Pax7-Flag expression. However, the proper control would be a Flag only expressing cell line. It is now difficult to assess what the contribution is of the Flag tag to the pull-down of the factor that are found. If the authors can somehow show that this is not an issue that would also be a solution.

Other points:

Please detail how are p-values in Fig. 5F calculated?

REVIEWERS' COMMENTS

Please see our responses to each comment below in bold text.

Reviewer #1 (Remarks to the Author):

All concerns addressed

Author response: We thank this Reviewer for supporting publication of our study.

Reviewer #2 (Remarks to the Author):

The authors have replied satisfactorily to my comments and made suitable changes. I therefore support the manuscript's publication in Nature Communications.

Author response: We thank this Reviewer for supporting publication of our study.

Reviewer #3 (Remarks to the Author):

I feel the authors have addressed most of my comments (i.e. more replicates for IP-MS, which is crucial). They have decided to tone down their conclusions with regard to the role of CTCF in Pax7-associated loops.

However, I would like to note that the IP-MS was not controlled as previously suggested. The control is a cell line without Pax7-Flag expression. However, the proper control would be a Flag only expressing cell line. It is now difficult to assess what the contribution is of the Flag tag to the pull-down of the factor that are found. If the authors can somehow show that this is not an issue that would also be a solution.

Author response: We outlined the control previously in our original manuscript. This was not a concern for other reviewers, and we have performed the controlled experiment in a standard way, with and without expression of the target protein, Flag-Pax7. This is how such a control is routinely performed given that the flag peptide is an 8-amino acid tag without an initiator methionine, so it is unlikely to be expressed or stable. Moreover, to our knowledge, this peptide is not known to interact with any specific proteins. We have performed many proteomic screens with other Flag-tagged proteins over the past 25 years and have never recovered the set of proteins that we identify in this study (or even a subset of these proteins). Repeating this experiment from scratch would require building the control cell line and re-doing everything at a cost of several

months. An alternate option would be to perform an IP-WB to check proteins we showed in Fig. 5b after overexpressing pax7-Flag and Flag-only constructs, but this would still take >1 month because we would need to generate a new Flag-only cell line. However, we do not believe that it will add any new information or alter the current list of proteins identified in our proteomic study for the reasons described above.

Other points:

Please detail how are p-values in Fig. 5F calculated?

We have added this information to the figure legend.